# Crystal structure of *Leishmania donovani* glucose 6-phosphate dehydrogenase reveals a unique N-terminal domain

Isabell Berneburg [1], Stefan Rahlfs[1], Katja Becker[1] & Karin Fritz-Wolf [1,2 ✉]

Since unicellular parasites highly depend on NADPH as a source for reducing equivalents, the pentose phosphate pathway, especially the first and rate-limiting NADPH-producing enzyme glucose 6-phosphate dehydrogenase (G6PD), is considered an excellent antitrypanosomatid drug target. Here we present the crystal structure of *Leishmania donovani* G6PD (*Ld*G6PD) elucidating the unique N-terminal domain of *Kinetoplastida* G6PDs. Our investigations on the function of the N-domain suggest its involvement in the formation of a tetramer that is completely different from related *Trypanosoma* G6PDs. Structural and functional investigations further provide interesting insights into the binding mode of *Ld*G6PD, following an ordered mechanism, which is confirmed by a G6P-induced domain shift and rotation of the helical N-domain. Taken together, these insights into *Ld*G6PD contribute to the understanding of G6PDs' molecular mechanisms and provide an excellent basis for further drug discovery approaches.

[1] Biochemistry and Molecular Biology, Interdisciplinary Research Center, Justus Liebig University, Giessen, Germany. [2] Max Planck Institute for Medical Research, Heidelberg, Germany. ✉email: Karin.Fritz-Wolf@ernaehrung.uni-giessen.de

The vector-borne infectious disease leishmaniasis is one of 20 poverty-related, neglected tropical diseases. It is caused by the protozoan parasite of the genus *Leishmania*, belonging to the *Trypanosomatidae* family, and infects 0.7 to one million people with 20,000 to 30,000 deaths annually in tropical, subtropical, and southern European regions. The species *Leishmania donovani* causes the most severe form, often fatal if untreated[1,2]. Currently, there are only a few drugs available for leishmaniasis treatment and after decades of use, resistant strains have emerged[3–5]. Therefore, new drug targets and anti-infective agents with new mechanisms of action are urgently required.

The enzymes of the pentose phosphate pathway (PPP) are considered promising drug targets for controlling trypanosomatids[5–7]. Throughout their life cycle, parasites must withstand the action of reactive oxygen and nitrogen species (ROS/RNS) produced by the host's immune system[8]. To counteract antioxidant defense mechanisms such as the trypanothione-dependent thiol system, which is crucial for redox balance in trypanosomatids, the parasites highly depend on NADPH as the primary electron source[9,10]. The PPP, which is divided into a unidirectional oxidative branch where NADPH is produced, and a non-oxidative branch that ends with ribose 5-phosphate, a crucial product for nucleotide biosynthesis, greatly contributes to the total NADPH pool. The housekeeping enzyme glucose 6-phosphate dehydrogenase (G6PD) (EC 1.1.1.49) is the first and rate-limiting enzyme of the PPP. It catalyzes the oxidation of glucose 6-phosphate (G6P) originating from glycolysis to 6-phospho-D-glucono-1,5-lactone (6PL), whereby the cofactor NADP$^+$ is reduced to NADPH. 6-phosphogluconate dehydrogenase (EC 1.1.1.44), the third enzyme of this oxidative branch generates a second NADPH molecule[10]. In trypanosomatids, the malic enzyme, which uses malate to produce pyruvate and NADPH, provides a second relevant source of NADPH[11,12]. However, RNAi downregulation of G6PD or 6-phosphogluconate dehydrogenase highly reduces the growth of *Trypanosoma brucei* bloodstream parasites, confirming the central role of these enzymes[7,13]. Additionally, some human steroids (e.g., dehydroepiandrosterone, epiandrosterone), have been shown to specifically inhibit *Trypanosoma* G6PDs, resulting in reduced parasite growth and a destroyed redox balance[7,13–15]. Since *Leishmania* promastigotes entering mammalian macrophages are exposed to enormous oxidative stress induced by enzymes such as NADPH oxidase or NO-synthase[16–18], an adequate NADPH pool is even more important for these parasites. Moreover,

the mode of action of commonly used antileishmanial drugs such as amphotericin B[19], sodium antimony gluconate[20], and miltefosine[21] is thought to be related to increased ROS production, and resistance to these drugs to an upregulation of the PPP[18]. These results clearly indicate a crucial role of the pentose phosphate pathway enzyme G6PD for *Leishmania* parasite survival and therefore its excellent suitability as a drug target, as previously shown for other unicellular parasites such as *Trypanosoma*[7,13–15] and *Plasmodium*[22–25].

In the present study, we describe the crystal structures of *Leishmania donovani* G6PD (*Ld*G6PD), both native and complexed with one or both substrates. The N-terminal domain, unique to *Kinetoplastida* G6PDs, is fully visible in our *Ld*G6PD wild-type (wt) crystal structures. Two *Ld*G6PD mutants provide interesting insights into the role of the additional N-terminal domain, which seems to be essential for tetramerization. Structural and kinetic investigations further elucidated the substrate binding mode of the enzyme, which seems to follow an ordered mechanism and is also evident through a domain shift due to G6P binding. The reported structural and functional characteristics of *Ld*G6PD provide an excellent basis for further structure-based inhibitor studies and drug optimization.

## Results

### Oligomerization and kinetic characterization of *Ld*G6PD wt, *Lm*G6PD wt, and mutants.
G6PD of *L. donovani* (*Ld*G6PD wt) and *L. major* (*Lm*G6PD), as well as two *Ld*G6PD mutants (*Ld*G6PD$^{C138S}$, *Ld*G6PD$^{60-562}$), were recombinantly produced (Fig. S1). To investigate the oligomerization behavior of *Leishmania* G6PDs, we performed size exclusion chromatography (SEC) under different conditions (see methods). The SEC profiles of native *Ld*G6PD wt revealed two stable peaks with elution patterns equivalent to tetrameric and dimeric forms of the enzyme, with the dimer peak dominating approximately twofold over the tetramer peak, independently of enzyme concentration (12–300 µM) (Fig. 1). The same elution profile could be observed for native *Lm*G6PD wt. Dimer and tetramer fractions remained in their original conformation when separately collected and reapplied to the SEC column (Fig. S2d). Steady-state kinetics revealed a specific activity of the *Ld*G6PD wt dimer of 200.0 ± 14.1 U mg$^{-1}$, with apparent $K_M$ values of 54.4 ± 2.8 µM for the substrate G6P and 15.9 ± 1.1 µM for the cosubstrate NADP$^+$. Compared to other

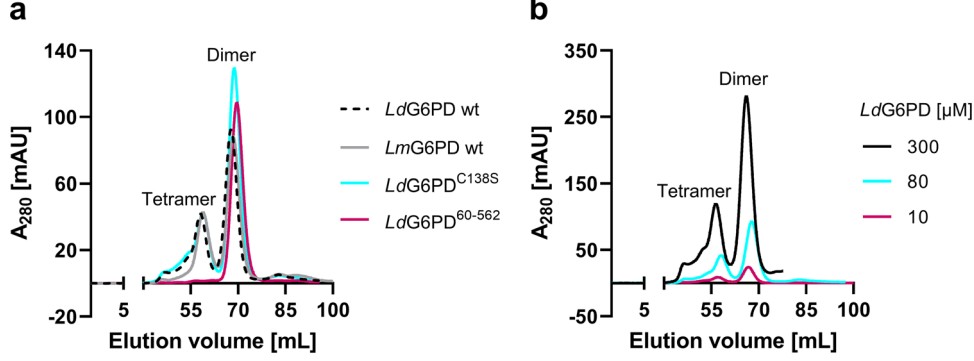

**Fig. 1 SEC analysis of recombinant *Ld*G6PD and *Lm*G6PD wt and of *Ld*G6PD mutants under native conditions.** Full-length G6PD was gel-filtrated on a HiLoad 16/60 Superdex 200 column pre-equilibrated in buffer A (500 mM NaCl, 50 mM Tris, pH 7.8). **a** The SEC profiles of *Ld*G6PD wt (66.7 kDa) (black dotted, retention volume (rv) 1. peak = 58 mL = 300 kDa, rv 2. peak = 68 mL = 133 kDa) and *Lm*G6PD wt (66.8 kDa) (gray, rv 1. peak = 59 mL = 276 kDa, rv 2. peak = 68 mL = 133 kDa) display two peaks with elution patterns equivalent to a dimer and a tetramer and remained stable in the *Ld*G6PD$^{C138S}$ mutant (66.7 kDa) (blue, rv 1. peak = 59 mL = 276 kDa, rv 2. peak = 68 mL = 133 kDa). The SEC profile of the *Ld*G6PD$^{60-562}$ mutant (60.3 kDa) displays one peak with elution patterns equivalent to a dimer (red, rv = 69 mL = 112 kDa). **b** SEC profiles of *Ld*G6PD wt under different enzyme concentrations, 300 µM (black, rv 1. peak = 56 mL = 341 kDa, 2. Peak = 66 mL = 156 kDa), 80 µM (blue, rv 1. peak = 58 mL = 300 kDa, rv 2. peak = 68 mL = 133 kDa), and 10 µM (red, rv 1. peak = 57 mL = 325 kDa, rv 2. peak = 67 mL = 144 kDa) display two peaks with stable elution patterns equivalent to a dimer and a tetramer. Representative chromatograms ($n ≥ 2$) are shown for each condition.

**Table 1 Comparative kinetic parameters of recombinant *Ld*G6PD wt and mutants, *Lm*G6PD, *Tc*G6PD, *Pf*GluPho, and *Hs*G6PD.**

| | G6P | | | | NADP+ | | | |
|---|---|---|---|---|---|---|---|---|
| | $V_{max}$ (U·mg⁻¹) | $K_M$ (μM) | $k_{cat}$ (s⁻¹) | $k_{cat}/K_M$ (μM⁻¹·s⁻¹) | $V_{max}$ (U·mg⁻¹) | $K_M$ (μM) | $k_{cat}$ (s⁻¹) | $k_{cat}/K_M$ (μM⁻¹·s⁻¹) |
| *Ld*G6PD$^{Dimer}$ | 200.0 ± 14.1 | 54.4 ± 2.8 | 222.5 ± 15.7 | 4.1 ± 0.2 | 204.7 ± 13.3 | 15.9 ± 1.1 | 227.7 ± 14.8 | 14.5 ± 1.6 |
| *Ld*G6PD$^{Tetramer}$ | 189.2 ± 7.7 | 75.4 ± 5.5 | 208.1 ± 5.1 | 2.8 ± 0.2 | 185.1 ± 12.9 | 14.2 ± 0.9 | 203.6 ± 13.6 | 14.9 ± 3.0 |
| *Ld*G6PD$^{C138S}$ | 228.1 ± 12.5 | 63.3 ± 2.3 | 253.6 ± 13.9 | 4.0 ± 0.1 | 224.7 ± 13.8 | 14.6 ± 1.6 | 249.8 ± 15.3 | 17.3 ± 0.8 |
| *Ld*G6PD$^{60-562}$ | 239.8 ± 10.6 | 65.4 ± 3.5 | 240.8 ± 10.6 | 3.7 ± 0.1 | 232.7 ± 18.7 | 14.3 ± 2.2 | 233.7 ± 18.8 | 17.0 ± 2.5 |
| *Lm*G6PD | 187.0 ± 2.8 | 57.9 ± 4.6 | 208.1 ± 3.1 | 3.6 ± 0.3 | 187.7 ± 1.6 | 11.4 ± 1.5 | 208.8 ± 1.8 | 17.9 ± 3.5 |
| *Tc*G6PD[29,39] | 99.4 ± 9.1 | 77 ± 20 | 62 ± 3 | 0.81 | 99.4 ± 9.1 | 16 ± 3 | 52 ± 2 | 3.3 |
| *Pf*GluPho[24] | 5.9 ± 1.0 | 20.3 ± 5.6 | 10.7 ± 1.8 | 0.53 | 5.9 ± 0.6 | 6.1 ± 1.3 | 10.1 ± 0.9 | 1.66 |
| *Hs*G6PD | 113.8 ± 6.7 | 70.1 ± 15.5 | 118.9 ± 7.0 | 1.7 ± 0.3 | 105.8 ± 7.2 | 22.9 ± 2.8 | 110.5 ± 7.6 | 5.0 ± 0.7 |

Values are expressed as mean ± SD from at least three independent determinations with different enzyme batches; each including at least three measurements.

G6PDs characterized so far, the specific activity is remarkably high, as is the overall catalytic efficiency (Table 1). In contrast, catalytic efficiency of the *Ld*G6PD tetramer was significantly lower as specific activity remained equal, but the apparent $K_M$ for G6P was about 40% higher (75.4 ± 5.5 μM) compared to the dimer fraction. Furthermore, the oligomerization pattern of *Ld*G6PD was neither affected by the presence of substrates nor by varying ionic strength (0.1–1 M NaCl) or pH values (4.8–7.8) (Fig. S2a, b). Nevertheless, very low pHs (4.5) reduced the specific activity by about 40% (114.5 + 8.2 U mg⁻¹). Moreover, neither reductive (5 mM DTT) nor oxidative conditions (2 mM $H_2O_2$) affected the elution patterns of *Ld*G6PD compared to native conditions (Fig. S2c), suggesting the formation of dimers and tetramers independent from redox status and intermolecular disulfide bonds. In line with that, the kinetic properties of the enzyme were not affected under the same conditions.

**Crystallization and structure determination of *Ld*G6PD.** We obtained monoclinic and orthorhombic crystals of *Ld*G6PD wt, *Ld*G6PD$^{C138S}$, and *Ld*G6PD$^{60-562}$ (Table 2). The crystals of the apoenzyme obeyed P12₁ symmetry with four monomers in the asymmetric unit (AU) and diffracted to 2.8 Å. The *Ld*G6PD wt and *Ld*G6PD$^{C138S}$ crystals in complex with and without its substrate G6P and/or its cofactor NADP(H) crystallized in space group C12₁ with two monomers in the AU. Crystals containing only NADP(H) diffracted to a resolution up to 1.7 Å, and crystals containing G6P or G6P and NADP(H) to 3.3 Å and 2.5 Å, respectively. The crystal of the truncated mutant *Ld*G6PD$^{60-562}$ obeyed C222 symmetry with one monomer in the AU and diffracted to 1.9 Å.

The structure of the apoenzyme was solved at 2.8 Å resolution via the molecular replacement method, using *Hs*G6PD (PDBID: 5UKW) as a search model. Diffraction data of all other crystals were phased via molecular replacement methods using the structure of the apoenzyme. All data collection and refinement statistics are summarized in Table 2. The electron density of the *Ld*G6PD core, comprising residues 5-470 and 483-552, is well defined in all our structures except two loops comprising residues D50-K64 and T471-D482. These loops have an increased B-factor; in some structures, they are not defined by the electron density (Table S1).

**Overall structure of *Ld*G6PD reveals a unique N-domain.** Similar to other G6PDs of this family, the *Ld*G6PD dimer (Fig. 2a) adopts the canonical G6PD fold with RMSD values for the dimer core domain of 2.6 Å and 2.1 Å to human (*Hs*G6PD; PDBID: 2BH9) and *Trypanosoma cruzi* G6PD (*Tc*G6PD; PDBID: 5AQ1)[26–29], respectively (Fig. S3). The *Ld*G6PD monomer contains three domains: an N-domain (Q5-K49), the NADP+

binding "Rossmann-like" domain (residues D50-I248), and a β + α domain (residues D249-K552). Each subunit has a discrete active site with an NADP+ and G6P binding domain. The latter is located between the "Rossmann-like" and the β + α domain. The N-domain consists of two α-helices comprising residues Q5-K49 and is linked to the "Rossmann-like" domain via a connecting loop (D50-K64).

The apo form crystallizes in space group P12₁ with two dimers (AB and CD) in the AU. Via P12₁ symmetry operation, a tetramer can be formed from each dimer (ABA'B' or CDC'D') (Figs. 2b, c, 3a). In contrast, the *Ld*G6PD wt and the *Ld*G6PD$^{C138S}$ mutant, complexed with NADP(H), G6P, or both substrates, adopt C12₁ space group symmetry with one dimer in the AU (AB). However, via crystallographic symmetry operation, the same kind of tetramer can be formed as with the apo form (Fig. 3b). Analysis of this tetramer, comprising two AUs, revealed two positions P1 and P2 of the N-domain of subunit B (N$^B$), which are related by a crystallographic symmetry operation and both located near the main domain of subunit B and B', respectively. In addition, the electron density of the loop D50-K64 (varies depending on the structure, Table S1), connecting the main domain of subunit B to N$^B$, is very low. Finally, we chose to associate P1 with subunit B and deposited the AU with this content in the PDB (Fig. 2a). The position of N-domain A (N$^A$) is unambiguous and described below.

As mentioned above, each N-domain comprises two α-helices α1 (~D11-R31) and α2 (~I36-K49) connected with a short loop (Figs. 2, 4). Tetramer contacts between the four subunits (A, B, A', B') exist mainly between the α-helices of the N-domains (Fig. 3). Substrate binding influences the contacts between the subunits and are described in more detail in the following section.

**Substrate binding leads to conformational changes and influences tetramer contacts.** Superimposing the dimeric apo *Ld*G6PD structure (PDBID: 7ZHT) with the NADP(H)-complexed *Ld*G6PD structures (PDBIDs: 7ZHU, 7ZHY) revealed RMSD values of 1 Å with 982 and 978 residues, respectively. Comparing the apo structure (Figs. 3a, 4a, b) with the structures complexed with G6P, NADP(H) or both substrates (Figs. 3b, 4c, d) reveals a rotation of the N-domain by about 45° upon ligand binding. In contrast to this striking realignment of the N-domains, only marginal differences in the position of the N-domains are evident between the ligand-bound structures.

In the apo structure, the N-domain of subunit A (N$^A$) interacts with the flexible loop (T471-D482) of both subunits (Fig. 3a). Within subunit A, the helical residues A26$^A$-E30$^A$ (α1) interact with loop residues Y472$^A$-Y476$^A$ (Fig. 4a). Residues E30$^A$-K32$^A$ are bound to the β-strand S442$^A$-A444$^A$ (β13). Furthermore, the R31$^A$ side chain interacts with E467$^A$ and D469$^A$ (β14). Upon

**Table 2 Data collection and refinement statistics.**

| | *Ld*G6PD wt | | | | *Ld*G6PD$^{60-562}$ | *Ld*G6PD$^{C138S}$ | |
|---|---|---|---|---|---|---|---|
| PDB code | 7ZHT | 7ZHU | 7ZHV | 7ZHW | 7ZHX | 7ZHY | 7ZHZ |
| Ligand | Apo | NADP(H) | G6P | G6P, NADP(H) | NADP(H) | NADP(H) | G6P, NADP(H) |
| *Data collection* | | | | | | | |
| Space group | P 1 2 1 | C 1 2 1 | C 1 2 1 | C 1 2 1 | C 2 2 2 | C 1 2 1 | C 1 2 1 |
| Cell dimensions | | | | | | | |
| $a, b, c$ (Å) | 117.7, 65.8, 189.2 | 223.2, 65.7, 119.3 | 218.7, 67.8, 120.8 | 215.8, 66.8, 121.0 | 75.8, 229.7, 83.6 | 223.2, 65.7, 119.9 | 218.2, 66.0, 120.0 |
| $\alpha, \beta, \gamma$ (°) | 90, 92.4, 90 | 90, 120.5, 90 | 90, 121.7, 90 | 90, 120.7, 90 | 90, 90, 90 | 90, 120.8, 90 | 90, 120.7, 90 |
| Resolution (Å) | 39.2-2.8 (2.9-2.8) | 48.1-1.7 (1.76-1.7) | 44.1-3.3 (3.2-3.1) | 49.3-3.3 (3.4-3.3) | 47.3-1.9 (2.0-1.9) | 47.95-2.0 (2.1-2.0) | 48.8-2.5 (2.6-2.5) |
| R-merge (%) | 11.4 (234.7) | 4.7 (132.0) | 11.5 (424.5) | 9.3 (158.3) | 5.6 (101.5) | 5.7 (96.1) | 5.8 (133.4) |
| I/σI | 8.6 (0.7) | 11.9 (0.6) | 6.9 (0.3) | 9.8 (1.3) | 18.9 (1.7) | 12.3 (1.2) | 12.0 (1.0) |
| Completeness (%) | 98.1 (97.6) | 97.8 (97.2) | 96.7 (95.6) | 99.1 (98.7) | 99.6 (99.2) | 99.6 (99.2) | 99.4 (98.8) |
| Redundancy | 2.8 (2.9) | 2.7 (2.7) | 2.9 (3.0) | 5.1 (5.2) | 5.6 (5.5) | 3.5 (3.2) | 3.5 (3.3) |
| Molecules per AU | 4 | 2 | 2 | 2 | 1 | 2 | 2 |
| Wilson B-factor | 73.9 | 34.0 | 133.9 | 124.8 | 33.4 | 40.3 | 75.6 |
| CC$_{1/2}$ (%) | 99.9 (44.6) | 99.9 (40.6) | 99.8 (25.7) | 99.9 (69.7) | 99.9 (81.2) | 99.9 (54.3) | 99.9 (52.2) |
| *Refinement* | | | | | | | |
| Resolution (Å) | 2.8 | 1.7 | 3.3 | 3.3 | 1.9 | 2.0 | 2.5 |
| No. reflections | 70481 | 159688 | 22260 | 22435 | 58616 | 102071 | 50909 |
| R$_{work}$/R$_{free}$ (%) | 24.4 (42.2)/ 30.4 (46.2) | 19.7 (39.4)/ 22.8 (41.6) | 25.8 (50.7)/ 31.4 (53.5) | 23.6 (44.5)/ 25.8 (44.5) | 18.2 (31.3)/ 21.0 (36.1) | 18.4 (37.6)/ 21.9 (40.2) | 24.9 (50.4)/ 29.8 (50.1) |
| No. atoms | | | | | | | |
| Proteins | 16027 | 8591 | 8342 | 8457 | 3826 | 8481 | 8344 |
| Ligands | 51 | 283 | 21 | 145 | 99 | 288 | 163 |
| Water | 26 | 890 | 4 | 1 | 381 | 753 | 25 |
| Protein residues | 2023 | 1077 | 1055 | 1069 | 481 | 1062 | 1055 |
| B-factors | | | | | | | |
| Proteins | 91.2 | 48.8 | 138.8 | 144.5 | 41.3 | 45.7 | 92.9 |
| Ligands | 98.1 | 52.9 | 161.4 | 169.2 | 43.5 | 51.2 | 111.0 |
| Water | 68.6 | 51.8 | 84.3 | 143.6 | 45.5 | 47.4 | 71.0 |
| Ramachandran plot (%) | | | | | | | |
| Favored (%) | 94.2 | 97.4 | 97.1 | 94.7 | 97.9 | 97.2 | 96.3 |
| R. m. s. deviations | | | | | | | |
| Bonds lengths | 0.009 | 0.007 | 0.01 | 0.003 | 0.009 | 0.019 | 0.003 |
| Bond angles | 1.71 | 0.90 | 1.18 | 0.62 | 0.98 | 1.08 | 0.67 |

Values in parentheses are for highest-resolution shell.

substrate binding the contacts to β13 are lost due to the rotation of N$^A$ and the resulting displacement of residues E30$^A$-K32$^A$ (Fig. 4c). However, the flexible loop (T471$^A$-D482$^A$) of subunit A also rearranges, thereby retaining T474$^A$ and Q27$^A$ binding (α1). Due to weak electron density, this interaction is not seen in structures with bound G6P (Table S1). As in a turnstile, the bonds between R31$^A$ and E467$^A$ and D469$^A$ remain intact because the side chains also rearrange.

Additionally, N$^A$ (apo structure) interacts with not only subunit A but also B; the N-terminus of helix α2$^A$ (Q38$^A$) is bonded to the flexible loop residues T471$^B$ and H473$^B$ (Figs. 4a, S4a). The N-domain rotation (N$^A$) due to substrate binding replaces this contact via hydrophobic interactions between D34$^A$-D35$^A$ and T474$^B$-R475$^B$ (Figs. 4c, S4b). The latter interaction is well defined by the electron density in almost all our structures with bound substrates except those with low resolution (3.3 Å) (Table S1).

As mentioned before, there are two positions P1 and P2 of N-domain N$^B$, which can be linked to the main domain of subunit B or B', respectively. Linking P1 to subunit B (Fig. 2b), results in no dimeric interactions between N$^B$ and subunit A, but in all our structures there are multiple contacts with the "Rossmann-like" domain of subunit B. In the apo form, α1$^B$ (A12$^B$-Q27$^B$) interacts via van der Waals forces with the helical

residues D117$^B$-W121$^B$ (α4$^B$) and E135$^B$-E141$^B$ (α5$^B$) (Fig. 4b). Upon substrate binding and the resulting rotation of N$^B$, the interactions are visible between the helical residues G19$^B$-A26$^B$ (α1$^B$), H123$^B$-K128$^B$ (α4$^B$) and E141$^B$ (α5$^B$) (Fig. 4d). Furthermore, Q27$^B$ is hydrogen-bonded to S134$^B$. Tetramer contacts between the four subunits (A, B, A', B') exist mainly between the α-helices of the N-domains (Figs. 2b and 3). Remarkably, there is one additional tetrameric contact, formed between N$^B$ (Q48$^B$–K49$^B$) and the flexible loop (Y476$^{B'}$-D477$^{B'}$) of subunit B' and vice versa.

If we connect position P2 of N-domain N$^B$ to subunit B, the above-described dimeric interactions with the "Rossmann-like" domain would take place between N$^B$ and subunit B' and would therefore be tetrameric interactions (Fig. 2c). However, the interactions between residues Q48$^B$-K49$^B$ and loop residues Y476$^B$-D477$^B$ would occur only within subunit B, rather than between two dimers as in the association of P1 to subunit B (Fig. 3b). Moreover, linking P2 to subunit B results in very fragile connections between the N- and the main domain. The connection would be via loop residues Q48$^B$-K64 $^B$ and Y476$^B$-D477$^B$ (Fig. 3b), which show increased B-factors or are disordered in some structures (Table S1). For these reasons, we consider it more likely that P1 and not P2 of N-domain N$^B$ is associated with subunit B.

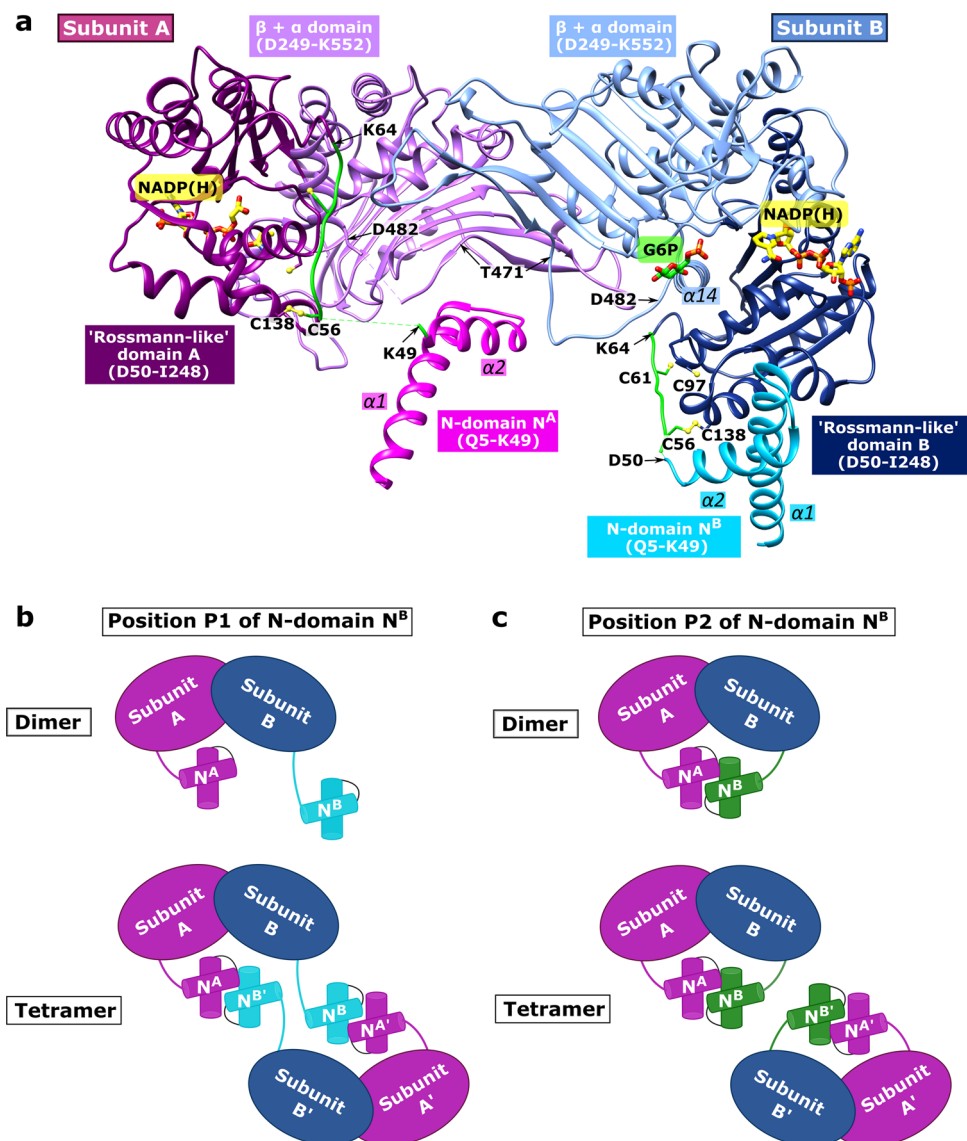

**Fig. 2 Overview of the *Ld*G6PD *wt* dimer and schematic model of the *Ld*G6PD oligomerization. a** *Ld*G6PD wt dimer in complex with NADP(H) (PDBID: 7ZHU) with position P1 of N-domain B (N$^B$) is shown in ribbon presentation. The β-α domain (D249-K552) of subunits A and B is colored light purple and light blue, respectively. The "Rossmann-like" domain (D50-I248) of subunits A and B is colored dark magenta and navy blue, respectively. The helical N-domain with helices α1 (~D11-R31) and α2 (~I36-K49) is colored magenta in subunit A (N$^A$) and cyan in subunit B (N$^B$). The connecting loop (residues D50-K64 in subunit A and B) between the N- and the "Rossmann-like" domain, is colored green. Regions with bad or missing electron density are shown in dotted lines (subunit A: D50-S55, Y476-R479; subunit B: R51-E54). Helix α14 as part of the G6P binding site, the flexible loop (T471-D482), and important cysteines (C56, C61, C97, C138, yellow) are labeled. NADP(H) (yellow) and G6P (green), and residues are shown as stick models. To mark the G6P binding pocket, G6P is taken from the superimposed structure with bound G6P (ribbons not shown, PDBID: 7ZHZ). **b** Schematic *Ld*G6PD dimer and tetramer involving position P1 of N-domain B (N$^B$). Subunit B is colored blue and N$^B$ of subunit B in position P1 is colored cyan. Subunit A and A' with N-domain A (N$^A$) and A' (N$^{A'}$) are colored magenta. **c** Schematic *Ld*G6PD dimer and tetramer involving position P2 of N-domain B (N$^B$). Subunit B is colored blue and N$^B$ of subunit B in position P2 is colored green. Subunit A and A' with N$^A$ and N$^{A'}$ are colored magenta (created with BioRender.com).

However, the described conformational changes upon NADP(H) binding did not extend to the NADP$^+$ binding pocket. Therefore, only minor differences occurred between the apo structure and structures complexed with NADP(H) or G6P with regard to the NADP$^+$ binding pocket. Residues forming the NADP$^+$ binding pocket are shown in Fig. 5a.

Significantly larger conformational changes than in the structures complexed with NADP(H) are observed after binding substrate G6P. Superimposition of the dimeric structures complexed with G6P (PDBID: 7ZHV) or with G6P and NADP(H) (PDBIDs: 7ZHW, 7ZHZ) with either the apo form (PDBID: 7ZHT; 965 residues) or the NADP(H)-bound form

(PDBID: 7ZHU; 1055 residues) revealed RMSD values of 1.6 and 1.8 Å, respectively. Moreover, the angle between the "Rossmann-like" domain and the β + α domain is reduced when G6P is bound (Fig. 6a). There are significant conformational differences in the vicinity of the G6P binding pocket in the structures with (PDBID: 7ZHZ) and without bound G6P (PDBIDs: 7ZHU, 7ZHY). Structural comparison of the monomers shows that regions I248–I260 (α9) and P481–F499 (α14) rearrange, with the main chain atoms shifting by about 1.5 Å and the Y251 side chain by about 3 Å (Fig. 6a, b). Furthermore, residues H250, K254, M256 (α9), H312 (α12), and Y484 (α14) adopt different side-chain conformations (Fig. 6b). Moreover, in the structures

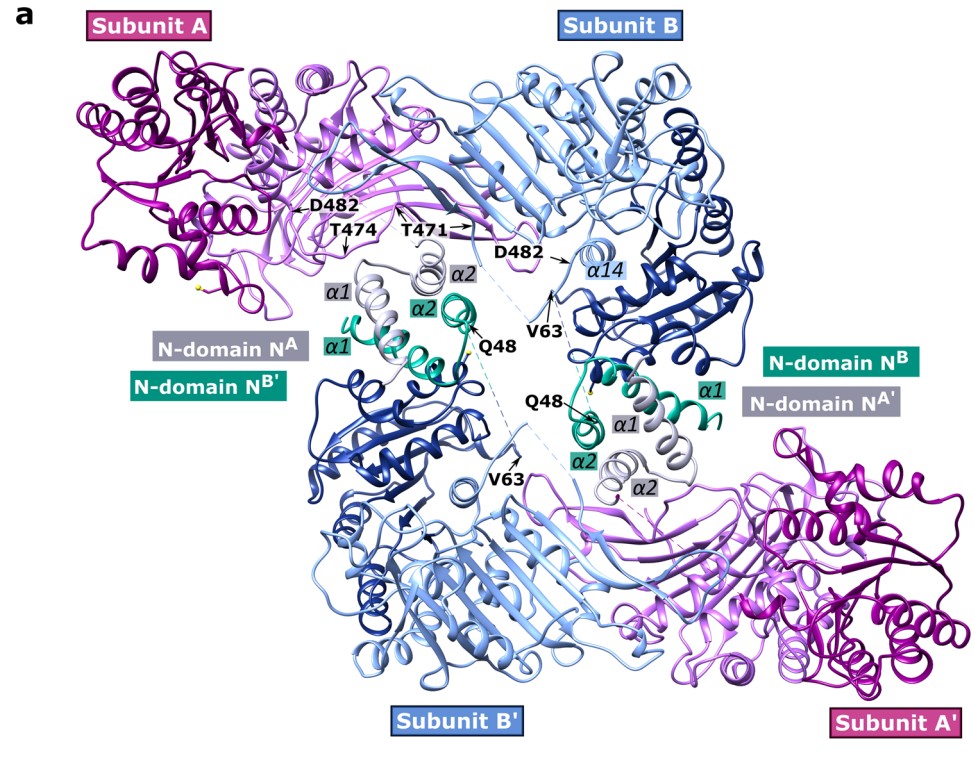

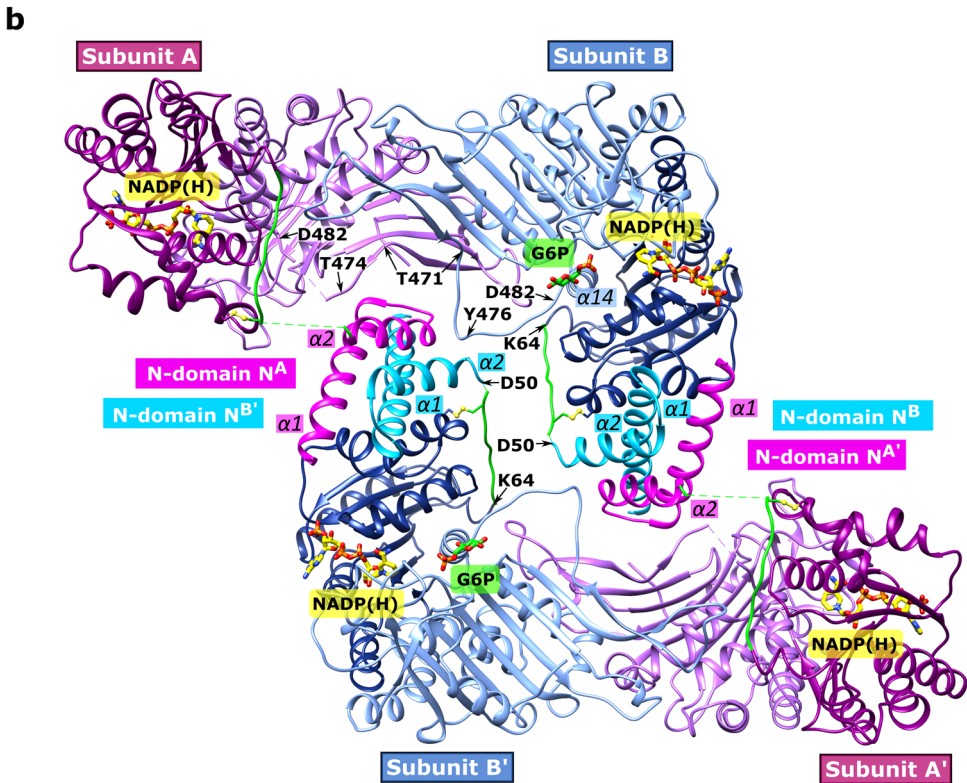

**Fig. 3 Overview of the *Ld*G6PD *wt* tetramer.** The tetramers of the apo **a** and the NADP(H)-complexed structures **b** are shown in the same orientation. The β-α domain (D249-K552) of subunits A/A′ and B/B′ is colored light purple and light blue, respectively. The "Rossmann-like" domain (D50-I248) of subunits A/A′ and B/B′ is colored dark magenta and navy blue, respectively. **a** apo structure. The helical N-terminal domain with α1 (~D11-R31) and α2 (~I36-K49) is colored gray in subunit A and A′ (N-domains N^A and N^A′) and colored dark cyan in subunit B and B′ (N-domains N^B and N^B′ with position P1). Dotted lines indicate regions with bad or missing electron density (subunit A/A′: G52-V63; subunit B/B′: K49-K62, T474-D477). **b** NADP(H)-complexed structure. N-domains N^A and N^A′ are colored magenta and N-domains N^B and N^B′ (position P1) are colored cyan. The connecting loop (D56-K64) is colored green, and dotted lines indicate regions with bad or missing electron density (subunit A/A′: D50-S55, Y476-R479; subunit B/B′: R51-E54). NADP(H) (yellow) and G6P (green) and residues are shown as sticks models. Important residues of the flexible loop (T471-D482) and the connected helix α14 (P481-L492) are directly or indirectly affected by the N-domains and marked with labels or arrows. The apo structure and the NADP(H)-complexed structure were used (PDBIDs: 7ZHT, 7ZHU), and G6P is taken from a superimposed structure with bound G6P (ribbon not shown, PDBID: 7ZHZ).

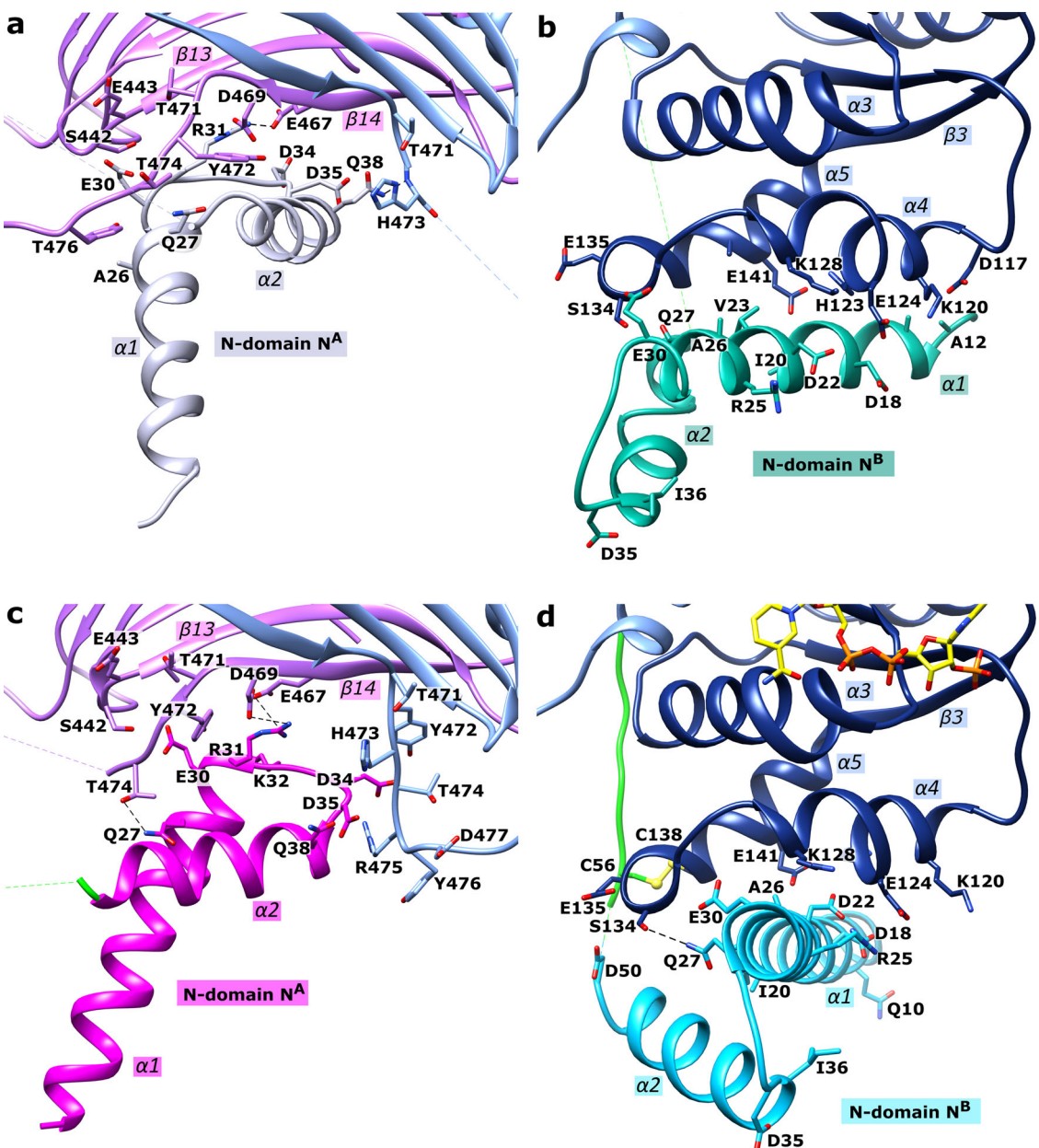

**Fig. 4 Close-up of the *Ld*G6PD wt N-domains.** The dimers of the apo and the NADP(H)-complexed structures are shown in the same orientation (PDBIDs: 7ZHT, 7ZHU). The β-α domain of subunits A or B are colored light purple and light blue, respectively. The "Rossmann-like" domain of subunit B is colored navy blue. The connecting loop (D50-K64) is colored green. **a** apo structure: N-domain A (N$^A$) is colored gray. **b** apo structure: N-domain B (N$^B$, position P1) is colored dark cyan. **c** NADP(H)-complexed structure: N-domain A (N$^A$) is colored magenta. **d** NADP(H)-complexed structure: N-domain B (N$^B$, position P1) is colored cyan; NADP(H) is colored yellow. Important residues are shown in stick models and hydrogen bonds are indicated with black dotted lines. Respective α-helices and β-sheets are numbered.

without bound G6P, the backbone atoms D482-A483 are located at the same position as the pyranose ring of G6P in G6P-complexed structures. Remarkably in all structures, G6P is bound only in subunit B of the G6PD dimer, although subunits A and B are essentially similar (RMSD ~ 0.7 Å). Minor shifts (~0.7 Å) occur for regions D117-E135 (α4) and cofactor NADP(H), and slightly higher shifts are observed for Y251 (1.5 Å) and residues P481-L492 (α14; 1.3 Å).

Comparison with homologous structures, in which the pyranose ring of G6P points towards NADP$^+$, revealed a corresponding G6P binding pocket in *Ld*G6PD, which is formed by residues K220, D249-Y251, F286-E288, K406, and K411-V413 (Fig. 5b). According to homologous structures, the pyranose ring

of G6P would be coordinated by residues K220, E288, I304, D307, H312, and K406, and the phosphate moiety would be bound by residues H250, Y251, K254, K411, and Q440.

However, in G6P-bound *Ld*G6PD structures, the position of G6P that best explains the electron density is different from that of homologous G6PD structures[26–29]. Compared to these structures, the G6P conformation is rotated by 180° and located in a cleft, formed by amino acids D482-A483, Q440, H250-Y251, and K254 (Fig. 5b). The pyranose ring is hydrogen-bonded to K254, A483, and Q440, which also interacts with the phosphate. Interestingly, in homologous structures residue K254 is used for phosphate binding. In our structures, the phosphate is bound by residues H250, Y251, and K411. Overall, the phosphate moiety

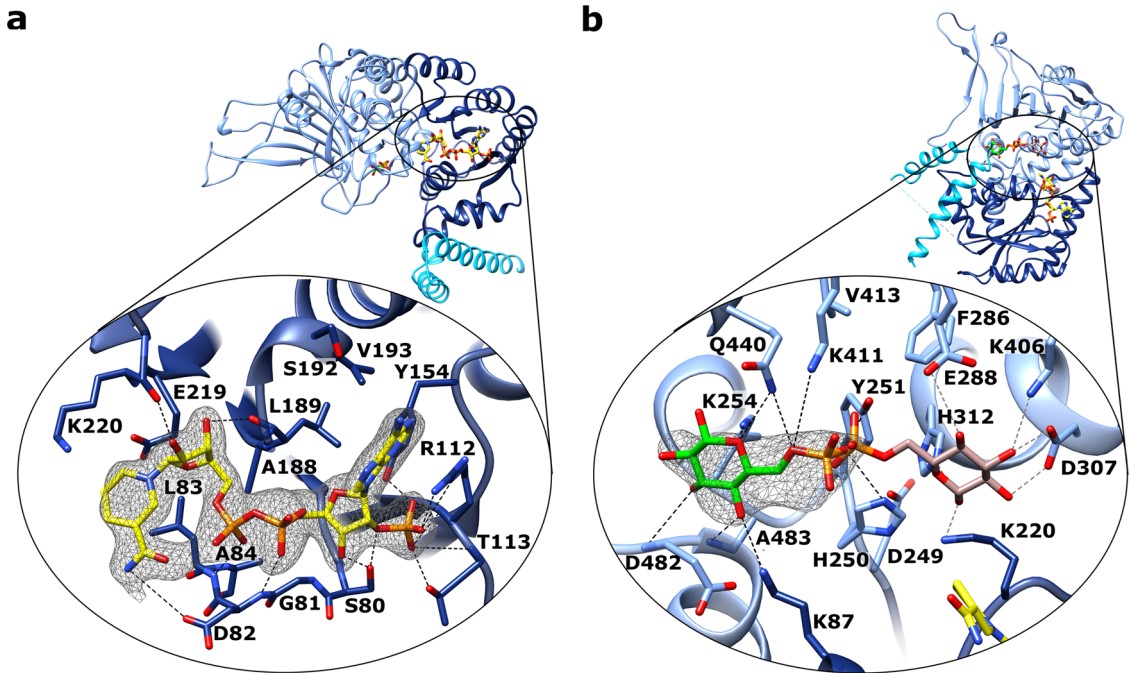

**Fig. 5 Active site close-up of *Ld*G6PD.** Ribbons and residues of the "Rossmann-like" domain are colored navy blue and of the β-α domain light blue. **a** Close-up of the NADP$^+$ binding pocket of the *Ld*G6PD structures (PDBID: 7ZHU) within the "Rossmann-like" domain. The NADP(H) moiety is colored yellow. Electron density map (F$_O$-F$_C$ polder omit map) contoured at 4.0 σ for NADP(H), is shown in black. **b** Closeup of the G6P binding pocket. G6P is shown in two conformations. In brown, the G6P conformation from the superimposed *Tc*G6PD structure (PDBID: 5AQ1) with the pyranose ring pointing towards the NADP(H) moiety (yellow). In green, the G6P conformation of the *Ld*G6PD structures (PDBID: 7ZHZ). Electron density map (F$_O$-F$_C$ polder omit map) contoured at 4.0 σ for G6P, is shown in black. Residues of the NADP$^+$ and the G6P binding pocket, are shown in stick models, and hydrogen bonds are indicated with black dotted lines or with gray dotted lines for the G6P moiety from the *Tc*G6PD structure.

shifts by only 1.7 Å compared to homologous G6PD structures (PDBIDs: 5AQ1, 6D24).

Taken together, binding NADP(H), G6P, or both results in a rotational shift of the two N-domains. This affects the interfaces between the N-domains and the core domain, particularly the contact region to the flexible loop T471-D482 so that residues of α14 (P481-L492) that participate in G6P binding are also affected. The NADP$^+$ binding site is less affected with only minor shifts. Furthermore, binding G6P induces a structural rearrangement of residues I248-I260 (α9) and P481-F499 (α14). Without bound G6P, the H250 and K254 side chains face away from the binding pocket, making the usual interaction with G6P impossible (Fig. 6a, b). Moreover, Y251 points into the G6P binding pocket, which would lead to clashes with the phosphate moiety of G6P in *Ld*G6PD and homologous structures (Fig. 6b, d).

**Truncation of residues 1–59 indicates the N-terminal extension is involved in *Ld*G6PD tetramerization.** To investigate a potential function of the helical N-domain conserved in *Kinetoplastida* G6PDs, we generated a truncation mutant lacking the N-domain that includes residues 1–59 (*Ld*G6PD$^{60–562}$). Moreover, the *Ld*G6PD wt structure, complexed with NADP(H), revealed a disulfide bond between C138 and C56; however, with minor conformational changes, an alternative disulfide bond between C138 and C61 or C56 and C97 is also possible (Fig. 2a). Therefore, we generated a second mutant by replacing C138 with a serine (*Ld*G6PD$^{C138S}$). We hypothesized that both mutations will impair enzyme activity due to truncation or altered orientation of the N-domain.

Comparing the *Ld*G6PD$^{C138S}$ structures to the wt structures revealed no major conformational changes (RMSD = 0.4 Å with 1056 residues). Accordingly, the mutation revealed no alternative disulfide bridge and the oligomerization profile did not

significantly differ from the wt (Fig. 1a). Although the $K_M$ of G6P increased by about 20%, indicating a lower affinity, the activity $V_{max}$ increased to the same extent so that catalytic efficiency remained stable (Table 1). For NADP$^+$, catalytic efficiency slightly increased from 14.5 ± 1.6 µM$^{-1}$ s$^{-1}$ to 17.3 ± 0.8 µM$^{-1}$ s$^{-1}$ due to the increased $V_{max}$ but stable $K_M$.

In contrast to the wt and *Ld*G6PD$^{C138S}$ crystals, which contain the physiological dimer in the AU, the truncation mutant *Ld*G6PD$^{60–562}$ crystallized in an orthorhombic space group (C222) with one molecule in the AU (Table 2). Consequently, the physiological dimer is formed by a crystallographic two-fold rotation axis. However, our usual tetramer (Fig. 3) cannot be constructed via crystallographic symmetry operations. Instead, a different tetramer is formed via C222 symmetry operations (Fig. 6c), which interestingly resembles the G6PD tetramer of related *Trypanosoma* species. Here, the *Ld*G6PD dimer–dimer interface is mainly formed by salt bridges (R268, K324, R326, D335, E336, and D393), keeping the dimers in a back-to-back orientation. Similar to our truncation mutant, *Trypanosoma* G6PDs have always been crystallized as truncated forms and therefore lack the first N-terminal 57 or 37 residues[28,29].

Superimposing the respective NADP(H)-bound *Ld*G6PD$^{60–562}$ (PDBID: 7ZHX) and *Ld*G6PD wt dimer (PDBID: 7ZHU) revealed an overall shift of 0.8 Å (475 residues) and small shifts (up to 1.4 Å) of the β + α and "Rossmann-like" domains, resulting in a slightly more open conformation of the truncation mutant. In accordance with these structural results and in contrast to the stable dimer-to-tetramer equilibrium of *Ld*G6PD wt, the tetramer was no longer visible in the SEC profiles of *Ld*G6PD$^{60–562}$ but only a single peak corresponding to the dimer form of the mutant (Fig. 1a). However, the crystallographic and SEC observations were not reflected by impaired catalytic efficiency of the enzyme (Table 1).

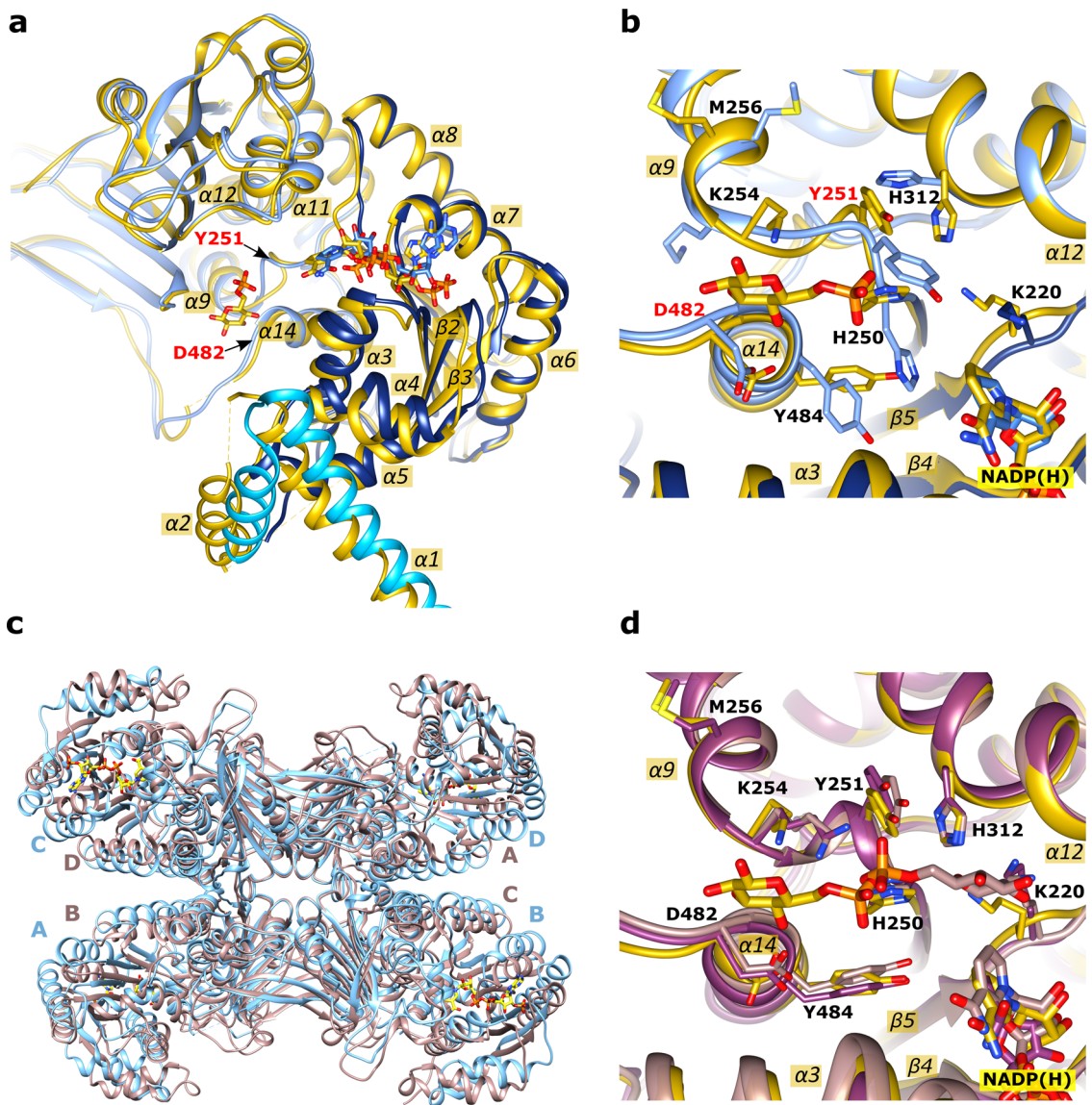

**Fig. 6 G6P-induced rearrangements of *Ld*G6PD wt and structural comparison with orthologues.** Superimposition of the NADP(H)-complexed structure (PDBID: 7ZHU; blue) with the structure complexed with G6P and NADP(H) (PDBID: 7ZHZ; yellow). Residue Y251 (red) marks the G6P-induced active site shift of the loop comprising residues H250-K254. **a** The superimposed dimers (only monomer B is shown) show the G6P-induced shift of the N- and the "Rossman-like" domain. Residue D482 (red) marks the end of the flexible loop (T471-D482), connected to α14 **b** Active site close-up. Here, monomers B of both structures are superimposed. The superimposition shows the different sidechain conformations of residues lining the G6P binding site and the rearrangement of region H250-K254. **c, d** Comparison with orthologues. **c** Superimposition of the *T. cruzi* G6PD tetramer with the *Ld*G6PD60-562 mutant. Both enzymes were crystallized lacking the helical N-domains. The apo-structure of the *Tc*G6PD tetramer (PDBID: 6D23) with subunits A-D is colored brown and that of the NADP(H)-complexed *Ld*G6PD60-562 tetramer (PDBID: 7ZHX) light blue. **d** Superimposed active sites of the *Ld*G6PD structure (with bound G6P and NADP(H); PDBID: 7ZHZ; yellow) with the *Tc*G6PD (with bound G6P; PDBID: 5AQ1; brown) and *Hs*G6PD structure (with bound NADP+; PDBID: 2BH9; purple). Figure **d** is shown in the same orientation as **b**. Respective α-helices and β-sheets are numbered.

**Mechanistic considerations of *Leishmania* G6PD.** In order to determine the catalytic binding mechanism of the substrates, G6P and NADP+ were titrated at various constant concentrations of the second substrate, and the relationship between substrate concentration and initial enzyme velocity was analyzed. For both substrates, the intersection of the Lineweaver–Burk plots left of the ordinate points towards a sequential order mechanism of substrate binding, in which both substrates must bind to the enzyme before product formation can occur[30] (Fig. 7a, b). Furthermore, the lines intercept below the abscissa, indicating that binding the first substrate promotes binding of the second substrate.

To differentiate between ordered and random sequential mechanisms of *Ld*G6PD, we conducted inhibition studies with the product inhibitor NADPH and the dead-end inhibitor glucosamine 6-phosphate (GlcN 6-P), an analog of G6P[31–33]. NADPH was found to be a competitive inhibitor with respect to NADP+ in *Ld*G6PD as indicated by the intersection on the x-axis (Fig. 7c) and increasing $K_M$ values with increasing inhibitor concentrations (Fig. S5a). Likewise, GlcN 6-P shows a competitive inhibition pattern towards G6P (Figs. 7e, S5c). Towards NADP+, GlcN 6-P had no inhibitory effect (Figs. 7d, S5b). In contrast, NADPH was also found to be a competitive inhibitor with respect to G6P (Figs. 7f, S5d).

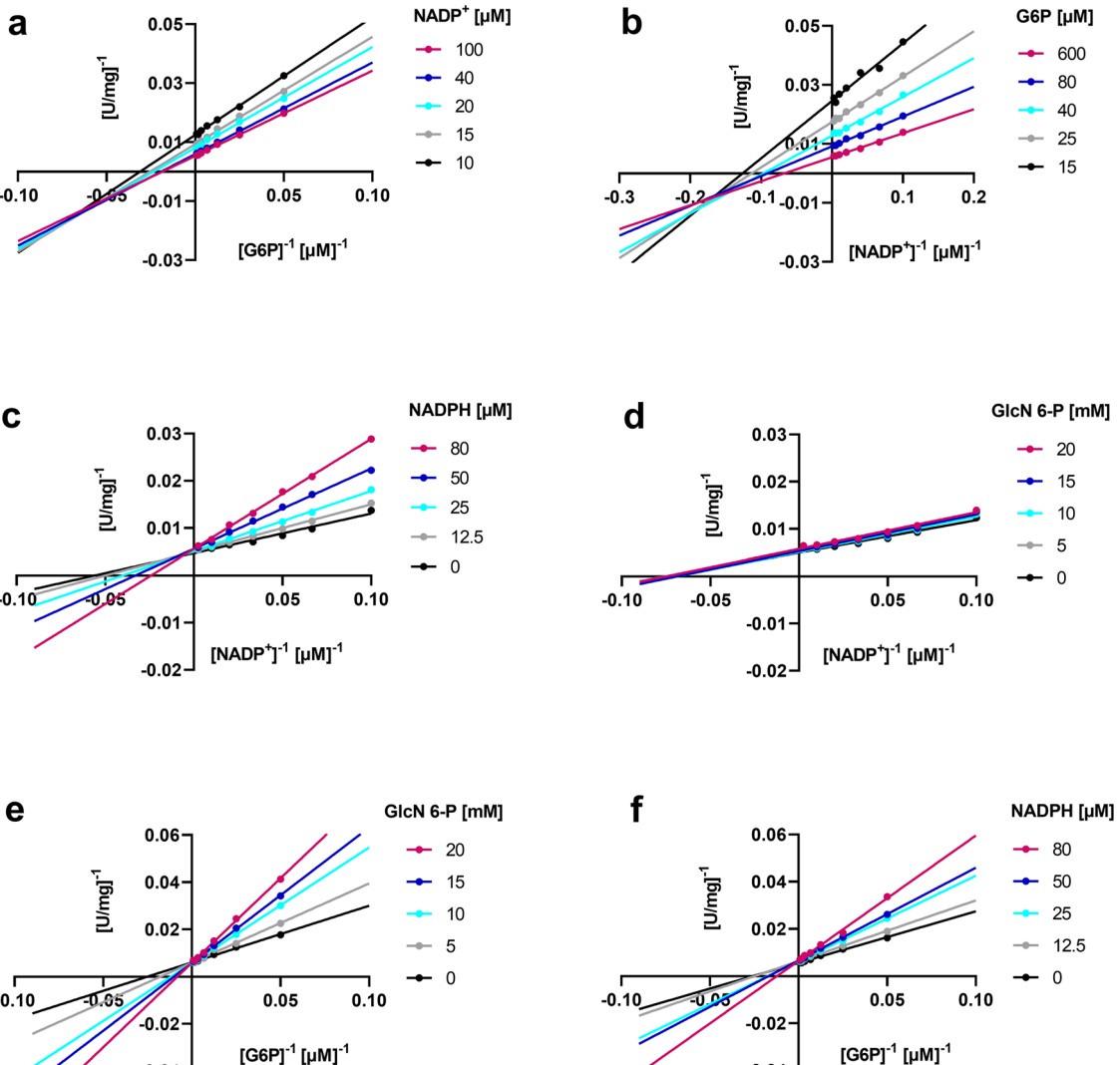

**Fig. 7 *Ld*G6PD's sequential binding mechanism for the substrates NADP$^+$ and G6P and inhibition by the product inhibitor NADPH and the dead-end inhibitor glucosamine 6-phosphate. a** NADP$^+$ and **b** G6P, as well as **c, f** NADPH and **d, e** glucosamine 6-phosphate (GlcN 6-P) were titrated at various constant concentrations of the substrate G6P or cosubstrate NADP$^+$. The primary double reciprocal plots of [U/100 mg]$^{-1}$ against [G6P]$^{-1}$ and [NADP$^+$]$^{-1}$, respectively, are shown. Plots **a** and **b** intercept left of the ordinate, indicating a sequential binding mode of the substrates. Plots **c**, **e**, and **f** intercept on the ordinate, indicating a competitive inhibition of NADPH against NADP$^+$ and G6P and of GlcN 6-P against G6P, but not against NADP$^+$. **d** Representative graphs from three independent replications are shown.

## Discussion

In this work, we present the crystal structure and biochemical characterization of *Leishmania donovani* G6PD. We were able to visualize the N-domain, which is unique in *Trypanosomatida*, providing interesting insights into its function and suggesting a central role in oligomerization. Additionally, we detected substrate-induced domain shifts extending to the active site of the enzyme. These conformational changes have never been observed in other G6PDs described so far. The crystallographic data also perfectly fit the kinetically determined binding mechanism of *Ld*G6PD. Finally, a much higher catalytic activity than homologous G6PDs supported these unique features, highlighting the central role of this enzyme in helping *Leishmania* parasites cope with oxidative stress.

Our kinetic investigations revealed a specific activity of about 200 U mg$^{-1}$ (Table 1) of recombinant *Ld*G6PD and *Lm*G6PD, with a 3–4-fold higher catalytic efficiency than G6PDs from other species. Moreover, SEC profiles of wt *Ld*G6PD and *Lm*G6PD revealed two peaks under native conditions, equivalent to a dimer

and a tetramer (Fig. 1a), whereas the tetramer fraction showed reduced catalytic efficiency compared to the dimer. Consequently, *Leishmania* G6PDs potentially exist in dynamic dimer–tetramer equilibrium, as with other G6PDs[26,29,33,34]. However, in contrast to homologous G6PDs[35–37], varying ionic strength, pH, enzyme concentration, or the presence of substrates did not affect this equilibrium (Fig. 1b, S2). The catalytic efficiency of *Ld*G6PD also remained unaffected under these conditions, except at very low pH (4.5), resulting in a 40% decrease in activity. However, *Leishmania* amastigotes in particular manage to keep the cytosolic pH largely neutral within the acidic parasitophorous vacuole[38], and enzyme activity could be restored when measured under neutral pH (7.5). Furthermore, reducing or oxidizing agents affected neither oligomerization nor the catalytic efficiency of *Ld*G6PD, whereas enzyme activity of the related *Tc*G6PD was almost abolished under such conditions[29,39]. The resistance of *Ld*G6PD to changing environmental conditions is probably an adaptation of *Leishmania* parasites, which are exposed to an extremely redox-active environment within the parasitophorous

vacuole in host macrophages and therefore rely on an adequate G6PD-derived NADPH pool.

Other trypanosomatid G6PDs such as from *Trypanosoma cruzi* or *Crithidia fasciculate* have been previously shown to form tetramers with a back-to-back orientation of the two dimers and a dimer–dimer interface predominantly formed by salt bridges[28,29]. Since the respective amino acids are conserved in *Kinetoplastida*, therefore also present in *Leishmania* G6PDs, all *Kinetoplastida* G6PDs have been postulated to likely form this kind of tetramers[28,29]. However, although the *Ld*G6PD SEC profiles showed an additional tetramer peak, varying ionic strength did not affect the dimer-tetramer equilibrium of *Ld*G6PD wt, suggesting a minor contribution of salt bridges in *Ld*G6PD tetramer formation.

*T. cruzi* G6PDs were always crystallized as truncated variants without the N-terminal domain[28,29]. If we also truncate the entire N-domain in *Ld*G6PD (*Ld*G6PD$^{60-562}$), we are able to construct via crystallographic symmetry operations the same tetramer arrangement as seen for *Tc*G6PD (Fig. 6c). However, the SEC profile of the truncation mutant exhibited a pure dimer, without the additional tetramer fraction (Fig. 1a). Our crystallization studies with full-length *Ld*G6PD always reveal a tetramer in which the N-domains form the dimer–dimer interface leading to a face-to-face orientation of the two dimers (Fig. 3). Therefore, we suggest that the N-domains are essential for forming the *Ld*G6PD tetramer.

In accordance with previous findings on *Tc*G6PD[28,29,39], truncation of the N-domain did not change the catalytic efficiency of the *Ld*G6PD dimer (Table 1). Only minor conformational changes of the core domains in the crystal structure of *Ld*G6PD$^{60-562}$ confirmed this when compared to the wt.

As mentioned earlier, we observed reduced catalytic efficiency of the tetramer with an increased $K_M$ for G6P but similar activity compared to the dimer. Since there are two possible positions of N-domain of subunit B (N$^B$), the choice of P1 or P2 determines dimeric and tetrameric contacts (Figs. 2b, c, 3). If position P1 belongs to monomer B (Fig. 2b) a tetramer is formed with a dimer–dimer interface between the N-domains. Another tetramer contact in this scenario exists between N$^B$ and the flexible loop (T471-D482) of subunit B' and vice versa between N-domain of subunit B' (N$^{B'}$) and subunit B (Fig. 3b). Conversely, if we would associate position P2 with monomer B (Fig. 2c), these contacts would be dimeric, and the dimer–dimer interface of the tetramer would reduce to the interaction between helix α1 of N$^B$ or N$^{B'}$ and the "Rossmann-like" domain (α4) of subunit B' or B, respectively. We assume that P1 of N-domain N$^B$ is connected to subunit B. Therefore, the N-domain of subunit B' (N$^{B'}$) influences the flexible loop (T471-D482) of subunit B and consequently affects the G6P binding site (α14; P481-L492) (Figs. 3, 6a, b). This tetrameric interaction perfectly explains the increased $K_M$ for G6P of the tetramer, whereas $K_M$ of NADP$^+$ and the overall activity remained unaffected compared to the dimer.

Finally, which function the tetramer has in vivo must still be clarified. Although the conditions we tested had no effect on oligomerization, there are many other factors that could influence the dimer–tetramer equilibrium, such as the product-to-substrate ratio, various other ROS, or protein interactions. For example, p53 is known to regulate human G6PD activity[40], and in trypanomatids, trypanothione reductase, a direct NADPH consumer, is thought to interact with G6PD[41]. Maybe the unique N-domain is also involved in such protein interactions, but this needs to be investigated in further studies.

The kinetic mechanism of various G6PDs has been intensively investigated and controversially discussed. While kinetic studies support an ordered sequential binding mechanism for many

G6PDs[31,42–44] a rapid equilibrium random sequential mechanism has been demonstrated for others such as *Hs*G6PD and *P. falciparum* G6PD[32,33]. To investigate the sequential binding mechanisms of *Ld*G6PD in more detail, we used product inhibition studies (Fig. 7). Under saturating conditions, NADPH acts as a competitive inhibitor against NADP$^+$, as does the dead-end inhibitor glucosamine 6-phosphate (GlcN 6-P) against G6P. Surprisingly, NADPH also acts as a competitive inhibitor against G6P when NADP$^+$ is in saturation, while GlcN 6-P does not compete with NADP$^+$ when G6P is in saturation. These observations suggest that NADP$^+$ can bind equally well to the free enzyme and the enzyme binary complex with already bound G6P, whereas G6P prefers binding to the free enzyme. This indicates that *Ld*G6PD follows an ordered sequential mechanism, where substrate binding and product release occur in a specifically ordered fashion. Our data even suggest that *Ld*G6PD follows a special ordered mechanism, the Theorell–Chance mechanism (Fig. 8). In this special case, substrate binding and product release occur so rapidly that the ternary complex EAB (*Ld*G6PD-G6P-NADP$^+$) exists for only a very short time and is therefore not kinetically significant[30,45]. For *Ld*G6PD this means that the substrate G6P (A) binds first to build EA (*Ld*G6PD-G6P) followed by binding of the cosubstrate NADP$^+$ (B) to form the product 6-phospho-D-glucono-1,5-lactone (6PL) (P), followed by formation of NADPH (Q). However, product formation (P, Q) occurs immediately after binding NADP$^+$ (B) so that the ternary complexes EAB and EPQ exist transiently but at such a low concentration that they cannot be detected via initial rate studies.

This fits well with our crystallographic observations of *Ld*G6PD. We observed conformational changes compared to the apo structure due to G6P binding alone (Fig. 6a, b). Within the G6P binding pocket, this conformational change was mainly visible through Y251 and surrounding residues (H250-Y254) (Fig. 6b). In structures without bound G6P, these residues (H250-Y254) always point to the G6P binding pocket, making catalysis impossible.

Interestingly, NADP(H) is always visible in both monomers of the dimer, while G6P is visible in monomer B only. Therefore, it seems that both monomers do not convert their substrates synchronously but rather delayed from each other, in which the occupancy of one binding site suppresses the catalysis of the other. The substrate-induced 45° rotation of the N-domains compared to the apo structure could be involved in this phenomenon (Figs. 3, 6). Therefore, the N-domain rotation changes the N-domain contacts to the main chain and the interaction between all subunits. Consequently, the contacts of N-domain A to the flexible loop B are reduced in the apo structure, influencing the G6P binding site (Figs. 3, 6, S4).

Our hypothesis that G6P binds first is further strengthened by the fact that we obtained crystals only by adding G6P (PDBID: 7ZHV) but not when crystallizing with NADP$^+$ alone. Therefore, structures complexed only with NADP$^+$, were always crystallized under catalysis. Accordingly, the bound NADP$^+$ could also be the product NADPH. Finally, we interpret the G6P-induced shift,

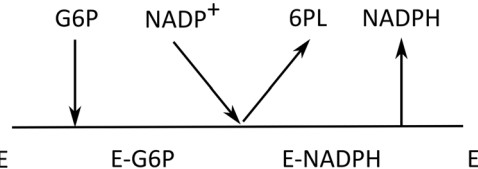

**Fig. 8 Proposed Theorell–Chance mechanism *Ld*G6PD.** G6P binds first to *Ld*G6PD followed by NADP$^+$ to generate the products 6-phospho-D-glucono-1,5-lactone (6PL) and NADPH without forming a significant ternary complex; E = enzyme *Ld*G6PD.

leading to a 45° rotation of the N-domain and a decreased angle between the β-α domain and the "Rossmann-like" domain, as a conformational change towards the catalytic form.

Interestingly, we observed a rearrangement of the G6P binding site upon G6P binding. In structures from other species, with or without bound G6P, the G6P binding site is similar to our structures with bound G6P, but not to the *Ld*G6PD structures without G6P (Fig. 6b, d).

However, the G6P conformation in the *Ld*G6PD structures is different from homologous G6PDs. Although the phosphate is similarly bound, the pyranose ring is rotated by 180° around the phosphate and therefore points in the opposite direction (Fig. 5b). In this conformation, the ring is located in a channel that extends to the center of the tetramer (Fig. 3). Consequently, electron transfer with NADP$^+$ and hence catalysis are impossible. We hypothesize that this conformation is a waiting position of G6P, whose position either results from the absence of NADP$^+$ (PDBID: 7ZHV) or is inhibited from catalysis by NADPH as a product inhibitor (PDBIDs: 7ZHW, 7ZHZ). However, as a feature of the postulated Theorell–Chance mechanism, the ternary complex exists only for a very short time, which could also explain why the true catalytic form is not visible in any of our structures.

The crystallographic and biochemical characterization of *Ld*G6PD presented in this study, revealed a trypanosomatid G6PD structure that contains the complete N-domain, unique to G6PDs of the *Kinetoplastida* family. Interestingly, our investigations on the function of the helical N-domain, indicated essentiality for *Ld*G6PD tetramerization and showed that the N-domain changes its position upon substrate binding. In contrast to homologous species, we observed G6P-dependent domain motions that kinetic studies supported, revealing an ordered mechanism with G6P binding first, followed by NADP$^+$. We believe that these insights from *Ld*G6PD will significantly contribute to the understanding of G6PDs' molecular mechanisms and will provide an excellent basis for further drug discovery approaches.

## Methods

**PCR amplification, cloning, and site-directed mutagenesis of G6PDs**. For heterologous overexpression in *E. coli*, the genes encoding wt *Ld*G6PD (UniProt acc. no. A2CIL3) and *Lm*G6PD (UniProt acc. no. Q4Q3K1) were ordered codon-optimized from BioCat GmbH (Heidelberg, Germany) and subcloned into the pET28a(+) vector using BamHI and HindIII restriction sites, thereby enabling N-terminally His-tagged proteins (Table S2).

To characterize the potential function of the N-terminal domain (covering the first 49 amino acids) two mutants were generated. C138 was replaced with serine (*Ld*G6PD$^{C138S}$) using the Q5® site-directed mutagenesis kit (New England Biolabs GmbH) with the construct pET28a(+)-*Ld*G6PD wt as a template. Furthermore, we generated a truncated construct lacking the first 59 N-terminal amino acids (*Ld*G6PD$^{60–562}$). PCR was performed using Q5 polymerase and the construct pET28a(+)-*Ld*G6PD wt used as a template. The PCR product was gel purified, digested with BamHI and HindIII and subsequently cloned into the BamHI-HindIII site of a pET28a(+) vector. All primers used in the present study are described in Table S3. The constructs were verified via sequencing from LGC genomics (Berlin, Germany) using standard primers within the T7 promotor and the T7 terminator sequence of the pET28a(+) vector.

**Expression and purification of His-tagged *Ld*G6PD wt, *Ld*G6PD$^{C138S}$, *Ld*G6PD$^{60–562}$, *Lm*G6PD wt**. Expression plasmids encoding for *Ld*G6PD and *Lm*G6PD wt and the corresponding mutants were transformed in *E. coli* C43 (DE3) cells using a pET28a(+) vector. Cells with the respective plasmid were grown at 37 °C in lysogeny broth medium containing 50 μg mL$^{-1}$ kanamycin. When the optical density at 600 nm reached ~0.5, heterologous overexpression was subsequently induced for 4 h by adding 0.5 mM IPTG. Cells were harvested via centrifugation (15 min, 12,000 × *g*, 4 °C), suspended in buffer A (500 mM NaCl, 50 mM Tris, pH 7.8), mixed with protease inhibitors (4 nM cystatin, 150 nM pepstatin, 100 μM PMSF) and stored at −20 °C until lysis. For cell lysis, lysozyme and DNaseI were added to the cell suspension and incubated for ~12 h at 4 °C, followed by three cycles of sonication and centrifugation (30 min, 25,000 × *g*, 4 °C). The supernatant was applied to a Ni-NTA column for IMAC purification, pre-

equilibrated with buffer A. Recombinant N-terminally His-tagged proteins were eluted with buffer A containing increasing (100–500 mM) imidazole concentrations. The fractions containing the recombinant protein were pooled and concentrated using an ultrafiltration unit (Vivaspin 20, 30-kDa filter cut-off; Sartorius, Göttingen, Germany). As an additional purification step and to determine the oligomerization state of the respective enzymes, size exclusion chromatography (SEC) was performed with an ÄKTA FPLC system, using a HiLoad 16/60 Superdex 200 column (GE Healthcare, Freiburg, Germany) pre-equilibrated with buffer A. To investigate the oligomerization behavior, SEC was performed under native, reductive (5 mM DTT), and oxidative (2 mM H$_2$O$_2$) conditions, as well as in the presence of substrates (100 μM G6P, NADP$^+$, G6P + NADP$^+$). Protein elution was detected at 280 nm and evaluated using the UNICORN software 7.2. The peaks of interest containing the recombinant protein were collected and concentrated and enzyme purity was assessed with Coomassie blue-stained SDS-PAGE gels (12% polyacrylamide). Enzyme concentration was determined by measuring the absorbance of the protein solution at 260 and 280 nm using the Eppendorf BioSpectrometer (Hamburg, Germany) followed by calculating the final concentration with the molecular weight and extinction coefficient of the respective enzymes (*Ld*G6PD wt/*Ld*G6PD$^{C138S}$: 66.7 kDa, 63,113 M$^{-1}$ cm$^{-1}$; *Ld*G6PD$^{60–562}$: 60.3 kDa, 61,623 M$^{-1}$ cm$^{-1}$; *Lm*G6PD wt: 66.8 kDa, 63,175 M$^{-1}$ cm$^{-1}$). After SEC, a final yield of 0.5–2 mg soluble, pure, and active G6PD per litre *E. coli* culture was achieved for all proteins tested in this study. The enzymes remained stable for at least 10 days when stored at 4 °C and 3 months when stored at −80 °C with the addition of 250 mM AmSO$_4$, as verified via enzyme activity measurements.

**Enzymatic characterization of *Ld*G6PD wt, *Ld*G6PD$^{C138S}$, *Ld*G6PD$^{60–562}$, *Lm*G6PD**. Using the spectrophotometer Evolution 300 (Thermo Scientific, Dreieich, Germany), G6PD activity was determined at 25 °C by following NADPH [$\varepsilon_{340} = 6{,}220$ M$^{-1}$ cm$^{-1}$] production at 340 nm according to Beutler[46]. All reactions were performed in buffer B (50 mM Tris, 3.3 mM MgCl$_2$, 0.005% Tween, 1 mg mL$^{-1}$ BSA, pH 7.5), whereas the enzymes were pre-diluted in buffer C (50 mM Tris, 0.005% Tween, 1 mg mL$^{-1}$ BSA, pH 7.5). The reaction mixture with a final volume of 500 μL contained 200 μM NADP$^+$, varying concentrations of the respective G6PDs (0.95–1.3 nM), and 800 μM G6P. To determine the apparent $K_M$ values and $V_{max}$, the substrate (1400–10 μM G6P) and the cosubstrate (500–2 μM NADP$^+$) were varied reciprocally. To determine the reaction mechanism, G6P was titrated at various concentrations of NADP$^+$ and vice versa. All measurements were performed at least in triplicate, and the apparent kinetic constants were calculated via nonlinear regression using the GraphPad Prism 9.0 software.

**Inhibition studies**. For product inhibition studies, the initial rates were measured for a series of NADPH concentrations (0–80 μM) with G6P in saturation (800 μM) and varying NADP$^+$ concentrations from 10 to 400 μM. Likewise, the experiment was carried out by varying the G6P concentration from 20 to 1200 μM with NADP$^+$ in saturation (200 μM) and again a series of NADPH concentrations (0–80 μM). In analogous fashion, 0–20 μM GlcN 6-P was used as an inhibitor covering the same combinations and ranges of substrate concentrations as used in the experiments with NADPH.

**Protein crystallization**. For crystallization of *Ld*G6PD wt and the respective mutants, *Ld*G6PD$^{C138S}$ and *Ld*G6PD$^{60–562}$, the enzymes were concentrated to ~10 mg mL$^{-1}$ in buffer A. Crystallization plates were prepared using a Digilab Honeybee 961 crystallization robot generating drops by mixing 0.2 μL reservoir solution with 0.2 μL protein solution. G6PD crystals were grown in a 96-well format with the sitting drop vapor diffusion technique. Initial crystallization conditions were identified within the screening of the JCSG Core I suite (Quiagen) and further optimized by screening around the initially identified conditions. Immediately after preparation, the plates were kept at 10 °C because crystal growth proceeded very fast, resulting in unstable, low-resolution crystals. Suitable crystals could only be obtained in complex with substrate and cosubstrate (1 mM each). *Ld*G6PD wt and *Ld*G6PD$^{C138S}$ crystallized best in a reservoir containing 10% PEG 3000 and 100–150 mM AmSO$_4$. The mutant *Ld*G6PD$^{60–562}$ crystallized in a reservoir containing 6% PEG 3000 and 200 mM AmCl$_2$ instead of AmSO$_4$.

We crystallized the *Ld*G6PD wt dimer and tetramer fraction separately. From the dimer fraction, we obtained crystals complexed with and without NADP(H), with G6P, and with both G6P and NADP(H). From the tetramer fraction, we used a crystal complexed with NADP(H) and G6P. From the dimer fraction of the *Ld*G6PD$^{C138S}$ mutant we got crystals complexed with NADP(H) and with both G6P and NADP(H). For the truncation mutant *Ld*G6PD$^{60–562}$ we obtained crystals in complex with NADP(H) only. A prerequisite for well-diffracting crystals (<2 Å) was the presence of both ligands, G6P, and NADP$^+$ in the crystallization buffer (Table 2, Table S1).

**Data collection and processing**. Diffraction data for all crystals were collected at beamline X10SA (Pilatus 6 M for the crystals of *Ld*G6PD wt and Eiger2 16 M for the crystals of *Ld*G6PD mutants) of the Swiss Light Source in Villigen, Switzerland at 100 K and processed with XDS[47]. Before data collection, crystals were soaked in mother liquor with a final concentration of 35% ethylene glycol.

The apoenzyme (PDBID: 7ZHT) crystallized in space group P121 with four monomers in the AU unit (Table 2). The wt enzyme in complex with G6P (PDBID: 7ZHV), the wt enzyme and the mutant *Ld*G6PD[C138S] in complex with NADP(H) (PDBIDs: 7ZHU, 7ZHY), as well as in complex with G6P and NADP(H) (PDBIDs: 7ZHW, 7ZHZ), crystallized in space group C121 with two monomers in the AU. All structures complexed with G6P or NADP(H) show C121 space group symmetry, but the structures with G6P have an a-axis shortened by about 5 Å. In addition to the two monoclinic crystal forms (P121, C121), we obtained orthorhombic crystals (C222) for the N-terminal truncation mutant *Ld*G6PD[60–562] (PDBID: 7ZHX) with one monomer in the AU.

**Structure determination and refinement**. A search model of *Ld*G6PD was generated via homology modeling with SWISS-MODEL[48] by using *Hs*G6PD (PDBID: 5UKW) as template. The modeled *Ld*G6PD comprises residues 70–549. In spite of the high sequence identity of 53% (Fig. S3), attempts to solve the structure of the apoenzyme using the canonical G6PD dimer were unsuccessful. Instead, a truncated version of the modeled *Ld*G6PD monomer was used, where loops on the surface, comprising residues 170–179, 359–376, 471–480, and 546–549, were deleted. The initial $R_{free}$ value after the first refinement was 37%. The AU of the final solution contained four monomers, which form two canonical G6PD dimers. In contrast to other G6PDs, *Ld*G6PD contains an additional N-terminal domain, which was only visible in one dimer of the apo structure. Diffraction data of all other crystals were phased via molecular replacement methods using the structure of the apoenzyme. During refinement, 8% of all reflections for the wt crystals complexed with NADP(H) were omitted and used for calculation of an $R_{free}$ value; for the other crystals, 10% of all reflections were used. The final statistics are shown in Table 2.

Interestingly, the apo structure and all structures complexed with G6P have much lower resolution (≥2.8 Å), than the structures in which only NADP(H) is bound (≤1.9 Å) (Table S2). The electron density of the *Ld*G6PD core comprising residues 5–470 and 483–552 is well defined in all our structures. In some structures, the electron density of the N-terminal domain (N-domain), covering the first 49 residues, is also well defined, although the temperature factors of the atoms are about 1.5 times higher than the average value of the core atoms. Two loops comprising residues D50-K64 and T471-D482 are defined by the electron density in some structures (Table S1), but the B-factor of these residues is at least twice the average B-factor.

The PHENIX program suite[49,50] served for reflection phasing and structure refinement. The interactive graphics program and Coot[51] were used for model building. Molecular graphics images were produced using the UCSF Chimera package[52].

**Statistics and reproducibility**. Data analysis was performed using GraphPad Prism Version 9.0 software. Data are presented as means ± SD. Reproducibility was confirmed by performing at least two independent determinations from different enzyme batches (for Fig. 1, Fig. S2) or three independent determinations with different enzyme batches, each including at least three (for Table 1) or two measurements (for Fig. 7 and Fig. S5) as described in the figure legend.

**Reporting summary**. Further information on research design is available in the Nature Portfolio Reporting Summary linked to this article.

## Data availability

Coordinates and measured reflection amplitudes have been deposited in the Worldwide Protein Data Bank RCSB PDB (http://pdb.org): code 7ZHT for *Ld*G6PD wt in its apo form; code 7ZHU for *Ld*G6PD wt in complex with NADP(H); code 7ZHV for *Ld*G6PD wt in complex with G6P; code 7ZHW for *Ld*G6PD wt in complex with NADP(H) and G6P; code 7ZHX for the truncation mutant *Ld*G6PD[60–562] in complex with NADP(H); code 7ZHY for mutant *Ld*G6PD[C138S] in complex with NADP(H) and 7ZHZ for the *Ld*G6PD[C138S] mutant in complex with NADP(H) and G6P. All source data behind the graphs and charts and unprocessed scans presented in the figures are presented in Supplementary Data 1.

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

## Acknowledgements

The authors wish to thank Michaela Stumpf for her excellent technical assistance. Furthermore, we would like to thank Ilme Schlichting and Miroslaw Tarnawski for the thoughtfully arranged data collection. Diffraction data were collected at beamline X10SA, Swiss Light Source, Paul Scherrer Institute, Villigen, Switzerland, and the authors thank the beamline staff for the excellent setup. The LOEWE Center DRUID (Projects B3 and E3) of the Hessian Excellence Program supported this work.

## Author contributions

Conceptualization (K.B., K.F.W., I.B.); Data curation (I.B., K.F.W.); Formal analysis (I.B., K.F.W.); Funding acquisition (K.B.); Investigation (I.B.); Methodology (I.B., K.F.W., S.R.); Resources (K.B., K.F.W.); Software (K.F.W.); Supervision (K.B., K.F.W.); Validation (K.F.W., I.B.); Visualization (I.B.); Roles/Writing—original draft (I.B.); Writing—review & editing (K.F.W., I.B., K.B., S.R.).

## Funding

## Competing interests

The authors declare no competing interests.
