## [Peer Review File · Communications Biology]

Reviewers' comments:

Reviewer #1 (Remarks to the Author):

The manuscript describes a set of crystallographic structures for *Leishmania donovani* G6PD showing for the first time its N-terminal domain and trying to explain how this domain would modulate the overall enzyme activity. This N-terminal domain is unique in kinetoplastids G6PDs and it is believed to regulate the enzyme activity (function) in the parasite. How kinetoplastid G6PDs are regulated remains an open issue. The structures and enzymatic data from this manuscript will represent a significant contribution to the field, if authors can address the following major concerns

Major issues:

1. Deletion of N-domain compromises the formation of LdG6PD tetramer in solution. This is clear in the size-exclusion chromatography experiments with LdG6PD60-562 (Fig. 1a). However, deletion of N-domain, or mutation of Cysteine-138 (C138S) doesn't affect the catalytic efficiency of the enzyme, K_{cat}/K_m values in table 1 for LdG6PD, LdG6PDC138S and LdG6PD60-562 are not significantly different. So, the biological function for the N-terminal domain of kinetoplastid G6PD's remains unsolved. Maybe, the N-terminal interacts with a different regulatory protein. That is not new for eukaryotic G6PDs, see the case of human G6PD and p53 interaction. (Jiang P, et al, p53 regulates biosynthesis through direct inactivation of glucose-6-phosphate dehydrogenase. *Nat Cell Biol.* 2011 Mar;13(3):310-6.)
2. The new tetrameric organization proposed for LdG6PD would benefit from additional biochemical data (cross-link assay, for instance), to confirm expected structure in solution.
3. In monomer B, poor electron density for the loop connecting the N-domain to Rossmann-like domain results in two possible orientations for the N-domain and makes the structural analysis very confusing.
4. It is not clear enough how the bind of substrates induces the 45° rotation of N-domain. What are the residues and forces involved in this movement?
5. The bind of G6P in a flipped orientation can also be interpreted as an crystallographic artefact.

Minor issues:

Line 457: "We crystallized the LdG6PD wt dimer and tetramer fraction separately. From the dimer fraction we obtained crystals complexed with and without NADP(H), with G6P, and with both G6P and NADP(H). From the tetramer fraction, we used a crystal complexed with NADP(H) and G6P. From the dimer fraction of the LdG6PDC138S mutant we got crystals complexed with NADP(H) and with both G6P and NADP(H). For the truncation mutant LdG6PD60-562 we obtained crystals in complex with NADP(H) only. A prerequisite for well-diffracting crystals ($< 2 \text{ \AA}$) was the presence of both ligands, G6P, and NADP+ in the crystallization buffer (Table 2, Table S1)." This would be better in the Results, not in Methods.

Line 422: "The enzymes remained stable for at least 10 days when stored at 4 °C and 3 months when stored at -80 °C with the addition of 250 mM AmSO₄, as verified via enzyme activity measurements." Same statement in line 394 and 395

Line 379: Please, provide the condon-optimized sequences in the supp. Material. This is an important information to allow data reproducibility.

Line 103: which residue atoms were used to calculate the RMSD? Be more specific.

Line 111: Replace connected by related

Line 142: Add an image (to supp. Info) of the electron density for these residues.

Line 103: indicate which residue atoms (C α , mainchain, all ...) were used to calculate the RMSD? Be more specific.

Line 465: Which Pilatus? Be more precise.

Line 472: Crystal with G6P have a larger (not shortene) a -axis than the APO 7zht. Verify.

Line 480: Indicate which residues were in the truncated search model.

Line 480: "The initial R_{free} value after the first refinement was 37%. The asymmetric unit of the final solution contained 4 monomers, which form two canonical G6PD dimers. In contrast to other G6PDs,

LdG6PD contains an additional N-terminal domain, which was only visible in one dimer of the apo structure." Move this to results.

Line 488: "...the apo structures". I counted only one APO structure, 7zht. Is that right?

Line 503: replace 7Z4X by 7zht

Line 504: replace 7Z4Y by 7zhy and 7Z4Z by 7zhz.

Reviewer #2 (Remarks to the Author):

In this manuscript the authors report the first structure of the glucose-6-phosphate dehydrogenase from *Leishmania* parasites, a major tropical disease for which only a few drugs are available and new resistant strains against those drugs are appearing. The authors solved 7 structures of LdG6PD and demonstrated the importance of quite unique N-terminal domain in forming tetramers completely different from those of G6PD enzymes from human and *Trypanosom cruzi*. Without the N-terminal domain, LdG6PD can form the canonical tetramers as in human and *Tc* G6PDs. Several G6P bound LdG6PD structures indicate that the bound GDP is flipped by 180 degrees around the phosphate positive relative to the conventional G6P binding pose, which is very intriguing. Upon binding of G6P the N-terminal two-helix domain rearranges their orientations relative to the core domain, an interesting observation that might be helpful in designing inhibitors unique to *Leishmania* G6PD without affecting human G6PD. The extensive enzyme kinetics with a number combinations of the substrates and cofactors largely corroborate the structural results, and the authors proposes a Theorell-Chance mechanism in which G6P binds first followed by NADP+ binding, leading to product 6PL and spent cofactor NADPH. The manuscript is overall well written with comprehensive descriptions of the novel G6PD tetramer structures including N-terminal domain shifts upon G6P binding. This is a significant step forward in understanding this class of G6PD and provides important insights into possible inhibitor designs against this neglected tropical disease. It is recommended for publication after the following issues are addressed.

Major points

1. G6P binding pose is flipped by 180 degrees compared with the homologue structures, for example *Tc*G6PD. This is a major departure from the conventional understanding of the G6PD catalysis where the pyranose ring is acted on by the enzyme leading to the production of 6-phosphogluconolactone, hence the pyranose ring needs to be close to the nicotinamide group of NADP(H).
 - a. Electron density of the unique, unusual G6P pose should be shown.
 - b. The authors describe that K254, Q440 and A483 make hydron bonds with the pyranose ring of G6P. To show the importance of this G6P waiting position, mutations need to be made on K254 and Q440 and enzyme kinetics measured. This will, hopefully, show the significance of the unusual G6P binding pose.
2. This also raises an important question whether the relationship between the flipped G6P binding and the significant structural changes observed in the N-terminal domains in the tetramer is physiologically relevant to the reaction mechanism. What if G6P sits in the "normal" position with the pyranose ring facing towards the nicotinamide ring of NADP(H)? Would the same set of substantial changes be observed in the N-terminal domain orientations relative the core domains?
3. Disulfide bond between C138 and C56 (line 201-202) in the N-terminal truncated mutant structure, LdG6PD(60-562). Normally readers would assume that this was observed in the N-terminal truncated LdG6PD which is mentioned in the sentence immediately preceding this sentence. How is it possible to see a disulfide bond with cysteine C56 which is not included in the construct? The authors need to include a panel or inset in Figure 2 that shows the electron density for the disulfide bond between C138 and C56. Or, is it possible that the disulfide bond is visible in the structure of the WT G6PD in complex with NADP(H) since the sentence refers to Figure 2? If so the authors need to clarify that and rephrase the paragraph.

Minor points:

1. "Pyran" vs. "pyranose". The authors use "pyran" throughout the manuscript referring to the pyranose ring of G6P. Normally a pyran ring has two double bonds in its 6-membered ring (5C+1O) and close to (but not perfectly) planar conformation while the 6 membered ring of G6P does not have double bonds, hence in the chair conformation. The authors should either clarify that they have indeed the pyran ring in G6P in their X-ray structure, or, use "pyranose" throughout the text.
2. Figure 2B. Important loops D50-K64 and T471-D482 are shown in both dimers, but according to Table S2, they are only visible in select monomers and dimers. While it is OK to show them in both, the authors should clarify that these were modeled in from the corresponding monomer/dimers.
3. "wt" on line 55: define "WT" in the first occurrence.
4. Lines 180-182 on the G6P interactions with G6PD. This sentence reads as if it is referring to the LdG6PD structure. But actually, it is about the G6P pose and interactions in the TcG6PD structure. This should be explicitly stated in the sentence.
5. Lines 217-218. This sentence is not precise, or almost incorrect. Actually, the WT tetramer coordinates are generated (constructed) by a different symmetry operation of totally different space groups, C121. As for 7ZHT, according to Table S1, it has already two dimers in the AU, so the tetramer might be exactly as shown in Figures 2 and 3; if not it can be generated via a P121 symmetry operation.
6. Figure S4: define the "nine" sequences. The figure shows only 5.

Finally, it would be interesting and instructive to discuss possible strategies for designing drugs against Leishmania parasites taking advantages of the unique N-terminal domain and it's interactions with the active sites across the loop T471-D482 or D50-K64.

ANSWERS TO THE COMMENTS OF EDITOR AND REVIEWERS

Manuscript ID: COMMSBIO-22-1550-T

Title: Chasing the Tail: Crystal Structure of *Leishmania donovani* Glucose 6-Phosphate Dehydrogenase Reveals a Unique N-terminal Domain

The authors wish to thank the editor and the reviewers for their constructive comments and for giving us the opportunity to resubmit our work. As delineated below we have taken into account all suggestions and carried out a number of changes. We modified (Fig. 2a, Fig. 3, Fig. 5, Table 2) or added additional figures/tables (Fig. 2 b+c, Fig. S4), added additional data (Table 1) and have rewritten several paragraphs. All changes to the manuscript are highlighted in blue. Paragraphs that have been removed are crossed out.

Reviewer comments:

Reviewer #1 (Remarks to the Author):

The manuscript describes a set of crystallographic structures for *Leishmania donovani* G6PD showing for the first time its N-terminal domain and trying to explain how this domain would modulate the overall enzyme activity. This N-terminal domain is unique in kinetoplastids G6PDs and it is believed to regulate the enzyme activity (function) in the parasite. How kinetoplastid G6PDs are regulated remains an open issue. The structures and enzymatic data from this manuscript will represent a significant contribution to the field, if authors can address the following major concerns.

Major issues:

1. Deletion of N-domain compromises the formation of LdG6PD tetramer in solution. This is clear in the size-exclusion chromatography experiments with LdG6PD60-562 (Fig. 1a). However, deletion of N-domain, or mutation of Cysteine-138 (C138S) doesn't affect the catalytic efficiency of the enzyme, Kcat/Km values in table 1 for LdG6PD, LdG6PDC138S and LdG6PD60-562 are not significantly different. So, the biological function for the N-terminal domain of kinetoplastid G6PD's remains unsolved. Maybe, the N-terminal interacts with a different regulatory protein. That is not new for eukaryotic G6PDs, see the case of human G6PD and p53 interaction. (Jiang P, et al, p53 regulates biosynthesis through direct inactivation of glucose-6-phosphate dehydrogenase. Nat Cell Biol. 2011 Mar;13(3):310-6.)

We thank the reviewer for these interesting comments. It is true that our studies have shown that the unique N-domain is not critical for enzyme activity. However, in our studies, we found by different methods that the N-domains are involved in tetramerization. Furthermore, we observed a reduced catalytic efficiency of the tetramer compared to the dimer, which was evident through an increased K_M for G6P by 40%. Our postulated LdG6PD tetramer supported this phenomenon since the G6P binding site (α 14; P481-L492) is affected in the tetrameric conformation through the interaction between N-domain B' and loop residues T471-D482 of subunit B.

Of course, the N-terminal domain could also play a crucial role in protein-interaction. Besides p53, trypanothione reductase is thought to interact with G6PD (Gosh *et al.*, G6PD and TR interaction protects *L. donovani* from metalloid mediated oxidative stress, Free Radic. Biol. Med., 2017). We enriched the manuscript with a small paragraph in the discussion that points that out (line 328-333):

'Finally, which function the tetramer has in vivo must still be clarified. Although the conditions we tested had no effect on oligomerization, there are many other factors that could influence the dimer-tetramer equilibrium, such as the product to substrate ratio, various other ROS, or protein interactions. For

example, p53 is known to regulate human G6PD activity⁴⁰, and in trypanomatids, trypanothione reductase, a direct NADPH-consumer, is thought to interact with G6PD⁴¹. Maybe the unique N-domain is also involved in such protein interactions, but this needs to be investigated in further studies.'

The function of the N-domain has been studied intensively for 15 years – until now it has remained unresolved. In this work, however, we were able to show for the first time the N-domain in atomic detail and that the entire domain moves upon substrate binding.

We agree with the reviewer that it would be very interesting to understand the function of the N-domain in detail. However, this would require intensive studies, including recombinant production of the interaction protein, production of substrates, kinetic analyses, SEC, HDX-MS, crystallization, production of further mutants etc.

2. The new tetrameric organization proposed for *LdG6PD* would benefit from additional biochemical data (cross-link assay, for instance), to confirm expected structure in solution.

Thank you very much for this comment. However, we are confident of our proposed tetrameric organization, because we have crystallized and analysed at least 30 structures (from different batches, different crystallization conditions, with and without ligands). In all of these structures, with the exception of the N-terminal deletion mutant (*LdG6PD*⁶⁰⁻⁵⁶²), which exhibited also a single dimer peak in the SEC profile, this type of tetramer is present. However, we agree that this tetrameric conformation is very interesting, which is why we would like to further investigate this tetramer and the involvement of the N-domains in another publication, as described under major point 1.

3. In monomer B, poor electron density for the loop connecting the N-domain to Rossmann-like domain results in two possible orientations for the N-domain and makes the structural analysis very confusing.

Thank you for this comment and we apologize for the inadequate presentation. We have rewritten several paragraphs (lines 110-120, 157-175, 311-322), optimized Fig. 2 a, and added a scheme (Fig. 2 b and c) for clarity.

Fig. 2 Overview of the *LdG6PD* wt dimer and schematic model of the *LdG6PD* oligomerization. **a** *LdG6PD* wt dimer in complex with NADP(H) (PDBID: 7ZHU) with position P1 of N-domain B (N^B) is shown in ribbon presentation. The β - α domain (D249-K552) of subunits A and B is colored light purple and light blue, respectively. The 'Rossmann-like' domain (D50-I248) of subunits A and B is colored dark magenta and navy blue, respectively. The helical N-domain with helices α 1 (~D11-R31) and α 2 (~I36-K49) is colored magenta in subunit A (N^A) and cyan in subunit B (N^B). The connecting loop (residues D50-K64 in subunit A and B) between the N- and the 'Rossmann-like' domain, is colored green. Regions with bad or missing electron density are shown in dotted lines (subunit A: D50-S55, Y476-R479; subunit B: R51-E54).

Helix α 14 as part of the G6P binding site, the flexible loop (T471-D482), and important cysteines (C56, C61, C97, C138, yellow) are labelled. NADP(H) (yellow) and G6P (green), and residues are shown as stick models. To mark the G6P binding pocket, G6P is taken from the superimposed structure with bound G6P (ribbons not shown, PDBID: 7ZHJ). **b** Schematic *LdG6PD* dimer and tetramer involving position P1 of N-domain B (N^B). Subunit B is colored blue and N^B of subunit B in position P1 is colored cyan. Subunit A and A' with N-domain A (N^A) and A' ($N^{A'}$) are colored magenta. **c** Schematic *LdG6PD* dimer and tetramer involving position P2 of N-domain B (N^B). Subunit B is colored blue and N^B of subunit B in position P2 is colored green. Subunit A and A' with N^A and $N^{A'}$ are colored magenta (created with BioRender.com).

4. It is not clear enough how the bind of substrates induces the 45° rotation of N-domain. What are the residues and forces involved in this movement?

This is correct and we are fully aware of this issue, but we cannot answer it properly at the moment. The rotation of the N-domain is clearly visible, but other involved regions show increased B-factors or are disordered. However, residues 471-481 of both subunits are rearranged with certainty upon substrate binding, but are not clearly defined by electron density in some structures (Table S2). We now have described in section 'Substrate binding leads to conformational changes and influences tetramer contacts' the involved residues in more detail (lines 137-166) and calculated polder maps of these regions and added a new figure (Fig. S4) to the supplements. Moreover, there are structural differences around the G6P binding site, described at the end of the same chapter (lines 197-209). We suppose that long range forces induce the rotation of the N-domain. Detailed mechanisms explaining the substrate induced rotation need to be investigated in further studies (for example: mutations of involved residues)

Fig. S4 Substrate-induced shift of the N-domains. The dimers of the apo and the NADP(H) complexed structures are shown in the same orientation (PDBIDs: 7ZHT, 7ZHU). The β - α domain of subunits A or B are colored light purple and light blue, respectively. Electron density map ($F_o - F_c$ polder omit map) contoured at 2.5σ around residues D34-L45 from N-domain A and residues T471-R475 of the loop region are shown in black. **a** apo structure: N-domain A (N^A) is colored grey. **b** NADP(H) complexed structure: N-domain A (N^A) is colored magenta. Important residues are shown in stick models. Respective α -helices and β -sheets are numbered.

5. The bind of G6P in a flipped orientation can also be interpreted as an crystallographic artefact.

Thank you for this comment. In the manuscript we showed only three structures with bound G6P (PDBIDs: 7ZHV, 7ZHW, 7ZHZ). However, we obtained many more crystals with bound G6P, all of which showed a similar orientation of the G6P, although the crystallization conditions were different. Moreover, the binding pocket of G6P does not form crystal contacts in our structures, so we assume that it is not a crystallographic artefact. In addition, we calculated a polder map and show the electron density in figure 5 b.

Fig. 5 Active site close-up of *LdG6PD*. Ribbons and residues of the ‘Rossmann-like’ domain are colored navy blue and of the β - α domain light blue. **a** Close-up of the NADP⁺ binding pocket of the *LdG6PD* structures (PDBID: 7ZHU) within the ‘Rossmann-like’ domain. The NADP(H) moiety is colored yellow. Electron density map ($F_o - F_c$ polder omit map) contoured at 4.0σ for NADP(H), is shown in black. **b** Closeup of the G6P binding pocket. G6P is shown in two conformations. In brown, the G6P conformation from the superimposed *TcG6PD* structure (PDBID: 5AQ1) with the pyranose ring pointing towards the NADP(H) moiety (yellow). In green, the G6P conformation of the *LdG6PD* structures (PDBID: 7ZHZ). Electron density map ($F_o - F_c$ polder omit map) contoured at 4.0σ for G6P, is shown in black. Residues of the NADP⁺ and the G6P binding pocket, are shown in stick models, and hydrogen bonds are indicated with black dotted lines or with grey dotted lines for the G6P moiety from the *TcG6PD* structure.

Minor issues:

1. Line 457: “We crystallized the LdG6PD wt dimer and tetramer fraction separately. From the dimer fraction we obtained crystals complexed with and without NADP(H), with G6P, and with both G6P and NADP(H). From the tetramer fraction, we used a crystal complexed with NADP(H) and G6P. From the dimer fraction of the LdG6PDC138S mutant we got crystals complexed with NADP(H) and with both G6P and NADP(H). For the truncation mutant LdG6PD60-562 we obtained crystals in complex with NADP(H) only. A prerequisite for well-diffracting crystals (< 2 Å) was the presence of both ligands, G6P, and NADP+ in the crystallization buffer (Table 2, Table S1).” This would be better in the Results, not in Methods.

We agree with the reviewer. Since, unfortunately, the journal allows only 5000 words for results and discussion we decided to keep this information in the method section.

2. Line 422: “The enzymes remained stable for at least 10 days when stored at 4 °C and 3 months when stored at -80 °C with the addition of 250 mM AmSO₄, as verified via enzyme activity measurements.” Same statement in line 394 and 395

Thank you very much for noticing that. We deleted the statement from line 392-395 (now lines 414-417).

3. Line 379: Please, provide the codon-optimized sequences in the supp. Material. This is an important information to allow data reproducibility.

Thank you for this objection. We added the codon-optimized sequences of *LdG6PD* and *LmG6PD* to the supplemental material as suggested (Table S2).

4. Line 103: which residue atoms were used to calculate the RMSD? Be more specific.

The superimposition was done with coot, using the routine SSM superpose. This is a topology-based alignment. Due to the limited space in the manuscript, we cannot list the residues in detail, but we rephrased this in the manuscript (lines 102-104).

‘Similar to other G6PDs of this family, the LdG6PD dimer (Fig. 2 a) adopts the canonical G6PD fold with RMSD values for the dimer core domain of 2.6 Å and 2.1 Å to human (HsG6PD; PDBID: 2BH9) and Trypanosoma cruzi G6PD (TcG6PD; PDBID: 5AQ1)26–29, respectively (Fig. S3).’

5. Line 111: Replace connected by related

Thank you for that. We have rewritten this paragraph, which is why this sentence does not longer exist.

6. Line 142: Add an image (to supp. Info) of the electron density for these residues.

We calculated polder maps and added additional figures (Fig. S4) to the supplemental material showing the electron density of the contact region between the N^A-domain and subunit B. The interactions between these residues change upon substrate binding, which is now also evident via the electron density map. See also major point 4.

7. Line 103: indicate which residue atoms (C α , mainchain, all ...) were used to calculate the RMSD? Be more specific.

Please see minor point 4.

8. Line 465: Which Pilatus? Be more precise.

We apologize for the missing information and have added this information to the manuscript (lines 486-488).

'Diffraction data for all crystals were collected at beamline X10SA (Pilatus 6M for the crystals of LdG6PD wt and Eiger2 16M for the crystals of LdG6PD mutants) of the Swiss Light Source in Villigen, Switzerland at 100 K and processed with XDS⁴⁷.'

9. Line 472: Crystal with G6P have a larger (not shortene) a -axis than the APO 7zht. Verify.

As shown in Table 2, in space group C121 the length of the a-axis in structures with bound G6P is at least 5 Å shorter than in structures with bound NADP(H). The apo structure crystalizes in space group P121. However, to be more precise, we rephrased the respective sentence (lines 493-495).

'All structures complexed with G6P or NADP(H) show C121 space group symmetry, but the structures with G6P have an a-axis shortened by about 5 Å.'

10. Line 480: Indicate which residues were in the truncated search model.

We thank the reviewer for this comment and checked the truncated model. Indeed, the number of the truncated residues was wrong. We have corrected this in the manuscript (lines 499-504).

'A search model of LdG6PD was generated via homology modeling with SWISS-MODEL⁴⁸ by using HsG6PD (PDBID: 5UKW) as at template. The modelled LdG6PD monomer comprises residues 70 to 549. In spite of the high sequence-identity of 53% (Fig. S3), attempts to solve the structure of the apoenzyme using the canonical G6PD dimer were unsuccessful. Instead, a truncated version of the modelled LdG6PD monomer was used, where loops on the surface, comprising residues 170-179, 359-376, 471-480 and 546-549, were deleted.'

11. Line 480: "The initial Rfree value after the first refinement was 37%. The asymmetric unit of the final solution contained 4 monomers, which form two canonical G6PD dimers. In contrast to other G6PDs, LdG6PD contains an additional N-terminal domain, which was only visible in one dimer of the apo structure." Move this to results.

We agree with the reviewer. Since, unfortunately, the journal allows only 5000 words for results and discussion we decided to keep this information in the method section.

12. Line 488: "...the apo structures". I counted only one APO structure, 7zht. Is that right?

Yes, that is right. Thank you for this comment. We rephrased the respective sentence (line 511).

13. Line 503: replace 7Z4X by 7zht

Thank you, this has been changed (line 526).

14. Line 504: replace 7Z4Y by 7zhy and 7Z4Z by 7zhz.

Thank you, this has been changed (line 527).

Reviewer #2 (Remarks to the Author):

In this manuscript the authors report the first structure of the glucose-6-phosphate dehydrogenase from *Leishmania* parasites, a major tropical disease for which only a few drugs are available and new resistant strains against those drugs are appearing. The authors solved 7 structures of LdG6PD and demonstrated the importance of quite unique N-terminal domain in forming tetramers completely different from those of G6PD enzymes from human and *Trypanosom cruzi*. Without the N-terminal domain, LdG6PD can form the canonical tetramers as in human and Tc G6PDs. Several G6P bound LdG6PD structures indicate that the bound GDP is flipped by 180 degrees around the phosphate positive relative to the conventional G6P binding pose, which is very intriguing. Upon binding of G6P the N-terminal two-helix domain rearranges their orientations relative to the core domain, an interesting observation that might be helpful in designing inhibitors unique to *Leishmania* G6PD without affecting human G6PD.

The extensive enzyme kinetics with a number combinations of the substrates and cofactors largely corroborate the structural results, and the authors proposes a Theorell-Chance mechanism in which G6P binds first followed by NADP⁺ binding, leading to product 6PL and spent cofactor NADPH. The manuscript is overall well written with comprehensive descriptions of the novel G6PD tetramer structures including N-terminal domain shifts upon G6P binding. This is a significant step forward in understanding this class of G6PD and provides important insights into possible inhibitor designs against this neglected tropical disease. It is recommended for publication after the following issues are addressed.

Major points:

- 1. G6P binding pose is flipped by 180 degrees compared with the homologue structures, for example TcG6PD. This is a major departure from the conventional understanding of the G6PD catalysis where the pyranose ring is acted on by the enzyme leading to the production of 6-phosphogluconolactone, hence the pyranose ring needs to be close to the nicotinamide group of NADP(H).**
 - a. Electron density of the unique, unusual G6P pose should be shown.**

Thank you for this objection. We calculated a polder map and show the electron density in figure 5 b. For consistency we included also a figure showing the electron density around NADP(H) (Fig. 5a).

At the beginning of the refinement, we tried to refine the *LdG6PD* wt structure with bound G6P (7ZHV) in its normal position, but then positive difference density ($> 3 \sigma$) could be seen in the channel and negative around the pyranose ring, thus we had to flip G6P. The unusual G6P position was also the best explanation for the electron density in the higher resolution structure of *LdG6PD*^{C138S} (7ZHZ).

Fig. 5 Active site close-up of *LdG6PD*. Ribbons and residues of the ‘Rossmann-like’ domain are colored navy blue and of the β - α domain light blue. **a** Close-up of the NADP⁺ binding pocket of the *LdG6PD* structures (PDBID: 7ZHU) within the ‘Rossmann-like’ domain. The NADP(H) moiety is colored yellow. Electron density map ($F_o - F_c$ polder omit map) contoured at 4.0σ for NADP(H), is shown in black. **b** Closeup of the G6P binding pocket. G6P is shown in two conformations. In brown, the G6P conformation from the superimposed *TcG6PD* structure (PDBID: 5AQ1) with the pyranose ring pointing towards the NADP(H) moiety (yellow). In green, the G6P conformation of the *LdG6PD* structures (PDBID: 7ZHZ). Electron density map ($F_o - F_c$ polder omit map) contoured at 4.0σ for G6P, is shown in black. Residues of the NADP⁺ and the G6P binding pocket, are shown in stick models, and hydrogen bonds are indicated with black dotted lines or with grey dotted lines for the G6P moiety from the *TcG6PD* structure.

b. The authors describe that K254, Q440 and A483 make hydrogen bonds with the pyranose ring of G6P. To show the importance of this G6P waiting position, mutations need to be made on K254 and Q440 and enzyme kinetics measured. This will, hopefully, show the significance of the unusual G6P binding pose.

We thank the reviewer for this suggestion. As mentioned in the manuscript (lines 197-209), comparison with homologous structures in which the pyranose ring of G6P points toward the NADP(H) moiety revealed that the conserved residues K254 and Q440 bind the phosphate group. Moreover, in human G6PD, mutation of the corresponding residue K254 showed, that this residue is essential for catalysis (Kotaka *et al.*, Structural studies of G6P and NADP⁺ binding to human G6PD. Acta Crystallogr., Sect. D: Biol. Crystallogr., 2005; Bautista *et al.*, Human G6PD. Lysine 205 is dispensable for substrate binding but essential for catalysis, FEBS Lett., 1995). The importance of a homologous glutamine residue to Q440 for catalysis has also been shown in several studies (Mercaldi *et al.*, The structure of a *T. cruzi* G6PD reveals differences from the mammalian enzyme. FEBS Lett., 2016; Kotaka *et al.*, Acta Crystallogr. Sect. D: Biol. Crystallogr., 2005). We therefore assume that mutation of these residues would result in an inactive enzyme. We know that region D482-A483 and Y251 block the usual G6P binding site when only NADP(H) is bound. However, when G6P is bound in the waiting position, the G6P binding site is open and catalysis could occur. Unfortunately, the exact function of this position is currently unclear and an interesting point for further studies, e.g. with G6P analogues and at transition states.

2. This also raises an important question whether the relationship between the flipped G6P binding and the significant structural changes observed in the N-terminal domains in the tetramer is physiologically relevant to the reaction mechanism. What if G6P sits in the “normal” position with the pyranose ring facing towards the nicotinamide ring of NADP(H)? Would the same set of substantial changes be observed in the N-terminal domain orientations relative the core domains?

We thank the reviewer for this interesting question but we cannot answer it properly at the moment. Binding of only NADP(H) induces a similar rearrangement of the N-domain as seen upon binding of G6P or both substrates. Therefore, we assume that G6P in its "normal" position would also lead to a rotation of the N-domain.

We added the following sentence (*Lines 137-140*):

‘Comparing the apo structure (Fig. 3 a, 4 a-b) with the structures complexed with G6P, NADP(H) or both substrates (Fig. 3 b, 4 c-d) reveals a rotation of the N-domain by about 45° upon ligand binding. In contrast to this striking realignment of the N-domains, only marginal differences in the position of the N-domains are evident between the ligand-bound structures.’

In all structures the rotation of the N-domain is clearly visible, but other involved regions show increased B-factors or are disordered. However, residues 471-481 of both subunits are rearranged with certainty upon substrate binding, but are not clearly defined by electron density in some structures (Table S2). We now have described more details in section ‘Substrate binding leads to conformational changes and influences tetramer contacts’. We calculated polder maps of these regions and added a new figure (Fig. S4) to the supplements. Moreover, there are structural differences around the G6P binding site, described at the end of the same chapter (lines 197-209). We suppose that long range forces induce the rotation of the N-domain. Detailed mechanisms explaining the substrate induced rotation need to be investigated in further studies (for example: mutations of involved residues).

Fig. S4 Substrate-induced shift of the N-domains. The dimers of the apo and the NADP(H) complexed structures are shown in the same orientation (PDBIDs: 7ZHT, 7ZHU). The β - α domain of subunits A or B are colored light purple and light blue, respectively. Electron density map ($F_o - F_c$ polder omit map) contoured at 2.5σ around residues D34-L45 from N-domain A and residues T471-R475 of the loop region are shown in black. **a** apo structure: N-domain A (N^A) is colored grey. **b** NADP(H) complexed structure: N-domain A (N^A) is colored magenta. Important residues are shown in stick models. Respective α -helices and β -sheets are numbered.

3. Disulfide bond between C138 and C56 (line 201-202) in the N-terminal truncated mutant structure, LdG6PD⁶⁰⁻⁵⁶². Normally readers would assume that this was observed in the N-terminal truncated LdG6PD which is mentioned in the sentence immediately preceding this sentence. How is it possible to see a disulfide bond with cysteine C56 which is not included in the construct? The authors need to include a panel or inset in Figure 2 that shows the electron density for the disulfide bond between C138 and C56. Or, is it possible that the disulfide bond is visible in the structure of the WT G6PD in complex with NADP(H) since the sentence refers to Figure 2? If so the authors need to clarify that and rephrase the paragraph.

We thank the reviewer for this observation and apologize for this error. We actually meant the *LdG6PD* wt structure with bound NADP(H). We have rephrased this sentence (lines 221-224).

‘Moreover, the LdG6PD wt structure, complexed with NADP(H), revealed a disulfide bond between C138 and C56; however, with minor conformational changes, an alternative disulfide bond between C138 and C61 or C56 and C97 is also possible (Fig. 2 a). Therefore, we generated a second mutant by replacing C138 with a serine (LdG6PD^{C138S}).’

Minor points:

1. “Pyran” vs. “pyranose”. The authors use “pyran” throughout the manuscript referring to the pyranose ring of G6P. Normally a pyran ring has two double bonds in its 6-membered ring (5C+1O) and close to (but not perfectly) planar conformation while the 6 membered ring of G6P does not have double bonds, hence in the chair conformation. The authors should either clarify that they have indeed the pyran ring in G6P in their X-ray structure, or, use “pyranose” throughout the text.

Thank you, you are right. We changed ‘pyran’ to ‘pyranose’.

2. Figure 2B. Important loops D50-K64 and T471-D482 are shown in both dimers, but according to Table S2, they are only visible in select monomers and dimers. While it is OK to show them in both, the authors should clarify that these were modeled in from the corresponding monomer/dimers.

We thank the reviewer for this remark. We suppose that the reviewer means Table S1 ‘Crystallographic Data’ and figures 3a and 3b ‘Overview of the *LdG6PD* wt tetramer’. It is correct that loops D50-K64 and T471-D482 are only visible in select monomers and dimers. The dotted lines connect the endpoints between the visible residues and indicate missing residues. We improved the figures and rephrased the legend.

Fig. 3 Overview of the *LdG6PD wt* tetramer. The tetramers of the apo **a** and the NADP(H)-complexed structures **b** are shown in the same orientation. The β - α domain (D249-K552) of subunits A/A' and B/B' is colored light purple and light blue, respectively. The 'Rossmann-like' domain (D50-I248) of subunits

A/A' and B/B' is colored dark magenta and navy blue, respectively. **a** apo structure. The helical N-terminal domain with $\alpha 1$ (~D11-R31) and $\alpha 2$ (~I36-K49) is colored grey in subunit A and A' (N-domains N^A and N^{A'}) and colored dark cyan in subunit B and B' (N-domains N^B and N^{B'} with position P1). Dotted lines indicate regions with bad or missing electron density (subunit A/A': G52-V63; subunit B/B': K49-K62, T474-D477). **b** NADP(H)-complexed structure. N-domains N^A and N^{A'} are colored magenta and N-domains N^B and N^{B'} (position P1) are colored cyan. The connecting loop (D56-K64) is colored green, and dotted lines indicate regions with bad or missing electron density (subunit A/A': D50-S55, Y476-R479; subunit B/B': R51-E54). NADP(H) (yellow) and G6P (green) and residues are shown as sticks models. Important residues of the flexible loop (T471-D482) and the connected helix $\alpha 14$ (P481-L492) are directly or indirectly affected by the N-domains and marked with labels or arrows. The apo structure and the NADP(H)-complexed structure were used (PDBIDs: 7ZHT, 7ZHU), and G6P is taken from a superimposed structure with bound G6P (ribbon not shown, PDBID: 7ZHZ).

3. "wt" on line 55: define "WT" in the first occurrence.

Thank you, this has been changed (now line 54).

4. Lines 180-182 on the G6P interactions with G6PD. This sentence reads as if it is referring to the LdG6PD structure. But actually, it is about the G6P pose and interactions in the TcG6PD structure. This should be explicitly stated in the sentence.

Thank you for this objection, the paragraph was indeed confusing and has now been rephrased (lines 197-201).

'Comparison with homologous structures, in which the pyranose ring of G6P points towards NADP+, revealed a corresponding G6P binding pocket in LdG6PD, which is formed by residues K220, D249-Y251, F286-E288, K406, and K411-V413 (Fig. 5 b). According to homologous structures, the pyranose ring of G6P would be coordinated by residues K220, E288, I304, D307, H312, and K406, and the phosphate moiety would be bound by residues H250, Y251, K254, K411 and Q440.'

5. Lines 217-218. This sentence is not precise, or almost incorrect. Actually, the WT tetramer coordinates are generated (constructed) by a different symmetry operation of totally different space groups, C121. As for 7ZHT, according to Table S1, it has already two dimers in the AU, so the tetramer might be exactly as shown in Figures 2 and 3; if not it can be generated via a P121 symmetry operation.

We thank the reviewer for this comment. The LdG6PD wt and the LdG6PD^{C138S} mutant complexed with NADP(H) and/or G6P adopt C121 space group symmetry, and via crystallographic symmetry operation the same kind of tetramer can be formed (see figure 3b). The apo form crystallizes in P121 with two dimers in the AU, however these two dimers do not form the biological tetramer (Fig. 3), they are quasi two half tetramers. Via P121 symmetry operation one can construct the same kind of tetramer (Fig. 3a) as constructed for the LdG6PD wt and the LdG6PD^{C138S} mutant (Fig. 3b), except the position of the N-domains. Regarding the N-terminal truncation mutant LdG6PD⁶⁰⁻⁵⁶² with one monomer in the AU and C222 space group symmetry, this type of tetramer cannot be formed via C222 symmetry operation, instead we can construct another type of tetramer (Fig 6c). We rephrased that paragraph (line 232-239) and the construction of the tetramer (line 110-120).

6. Figure S4: define the “nine” sequences. The figure shows only 5.

Thank you very much for noticing that. That was a mistake. We rephrased the respective sentence in the figure legend. Figure S4 is now Figure S3.

‘Strictly conserved residues are highlighted with black background, while highly similar residues, with similar physicochemical properties in at least four out of the five sequences are shown in bold letters.’

7. Finally, it would be interesting and instructive to discuss possible strategies for designing drugs against Leishmania parasites taking advantages of the unique N-terminal domain and it’s interactions with the active sites across the loop T471-D482 or D50-K64.

This is indeed interesting, which is why we already started with docking studies and inhibitor testing. However, such intensive studies would exceed the scope of this manuscript, which is why we decided to address this in another publication.

Reviewers' comments:

Reviewer #1 (Remarks to the Author):

Despite answering all raised issues, the reported data is not robust enough to rebut the argument that the tetrameric arrangement proposed for LdG6PDH could be a crystallographic artefact.

The electron density map presented in Fig5. B doesn't have enough resolution to allow the accommodation of the G6P in this unusual, flipped orientation. Maps from higher resolution structure would be necessary to convince article readers that the flipped G6P is correct.

The reported difference (of 40%) between Km-G6P values for LdG6PDH dimer and tetramer is too small to believe that changes between dimer and tetramer would have any biological significance for the parasite growth or viability.

The new finds based on Leishmania G6PDH structures go against what was established based on T. cruzi and Human G6PDH structures. So, these finds require more and better data to be trustable.

Reviewer #2 (Remarks to the Author):

The authors have addressed all the points raised by this reviewer. The revised manuscript is recommended for publication in Communications Biology.

ANSWERS TO THE COMMENTS OF THE REVIEWERS

Manuscript ID: COMMSBIO-22-1550-T

Title: Chasing the Tail: Crystal Structure of *Leishmania donovani* Glucose 6-Phosphate Dehydrogenase Reveals a Unique N-terminal Domain

The authors wish to thank the reviewers for their constructive comments and for giving us the opportunity to resubmit our work. We updated the discussion to clarify the key statements of our manuscript. All changes to the manuscript are highlighted in yellow.

Reviewer comments:

Reviewer #1 (Remarks to the Author):

Despite answering all raised issues, the reported data is not robust enough to rebut the argument that the tetrameric arrangement proposed for LdG6PDH could be a crystallographic artefact. The electron density map presented in Fig5. B doesn't have enough resolution to allow the accommodation of the G6P in this unusual, flipped orientation. Maps from higher resolution structure would be necessary to convince article readers that the flipped G6P is correct. The reported difference (of 40%) between Km-G6P values for LdG6PDH dimer and tetramer is too small to believe that changes between dimer and tetramer would have any biological significance for the parasite growth or viability. The new finds based on Leishmania G6PDH structures go against what was established based on T. cruzi and Human G6PDH structures. So, these finds require more and better data to be trustable.

To date, there are only 29 G6PD X-ray structures from two prokaryotic and three eukaryotic species (*Homo sapiens*, *Trypanosoma cruzi* and *Kluyveromyces lactis*) available. The three previous published *T. cruzi* G6PD and the three tetrameric human G6PD structures show a maximal resolution of 2.65 Å. In our current study, we have measured more than 30 LdG6PD crystals with and without substrates up to a resolution of 1.7 Å. Furthermore, our studies have been carried out with wild type and mutated enzyme.

Previous studies on *T. cruzi* G6PD have postulated a tetramer with a back-to-back orientation of the two dimers. This back-to-back arrangement was also found in the human G6PD tetramer, but here the dimers are oriented differently than in *T. cruzi*. Therefore, the tetramers of the two species are not identical. This is not surprising, since trypanosomatid G6PDs, such as from *Trypanosoma* and *Leishmania*, contain a unique N-terminal domain, which is not present in the human enzyme. Previous crystallization studies of *T. cruzi* G6PD were performed using only a truncated mutant lacking the N-terminal domain. If we also truncate the N-terminal domain in LdG6PD, we are able to construct via crystallographic symmetry operations the same tetramer arrangement as seen for *T. cruzi* G6PD. However, the tetramer peak is then no longer visible in the SEC profiles of the truncated LdG6PD, in contrast to full-length LdG6PD where a dimer and a tetramer peak are observed in the SEC profiles. In our studies crystallization of full-length LdG6PD always reveals a face-to-face tetramer in which the N-domains are involved in tetramer formation.

It was previously postulated, that all *Kinetoplastida* G6PDs would form a tetramer via conserved salt bridges. In contrast to the studies on *T. cruzi* G6PD, our study proved that the dimer-tetramer equilibrium of untruncated LdG6PD is independent of ionic strength, suggesting a minor contribution of salt bridges in LdG6PD tetramer formation. Therefore, we

suppose that the *T. cruzi* back-to-back tetramer is an artefact, due to the truncation of the N-terminal domain.

For the above reasons, it cannot be assumed that previous knowledge about *Trypanosoma* G6PDs is '*established*' and could therefore be used as a standard or template. Despite intensive studies over the last 15 years, the role of the N-domain has not yet been definitively clarified. Our studies provide a first and very important approach to this question. Similarly, all studies in the field of G6PD also covered only the methods we used and no studies have yet elucidated the physiological significance of the postulated dimer and tetramer conformations of G6PDs.

Finally, we would like to emphasize that the tetrameric arrangement of *LdG6PD* is not the main statement of our manuscript. In our work, we present for the first time a G6PD structure from the *Trypanosomatidae* family that contains the complete N-domain. Moreover, we show that this domain changes its position relative to the main domain upon substrate binding, and together with our kinetic studies, we gained new insights into the binding mode of *LdG6PD*.

Although we agree, that investigating the function of the tetramer in solution would be very interesting, this would certainly take at least half a year and is beyond the scope of this publication.

Reviewer #2 (Remarks to the Author):

The authors have addressed all the points raised by this reviewer. The revised manuscript is recommended for publication in Communications Biology.

We are very pleased that we were able to address all of the concerns and that the reviewer fully recommends publication.

ANSWERS TO THE COMMENTS OF EDITOR AND REVIEWERS

1. Revision – 08/23/2022

Manuscript ID: COMMSBIO-22-1550-T

Title: Chasing the Tail: Crystal Structure of *Leishmania donovani* Glucose 6-Phosphate Dehydrogenase Reveals a Unique N-terminal Domain

The authors wish to thank the editor and the reviewers for their constructive comments and for giving us the opportunity to resubmit our work. As delineated below we have taken into account all suggestions and carried out a number of changes. We modified (Fig. 2a, Fig. 3, Fig. 5, Table 2) or added additional figures/tables (Fig. 2 b+c, Fig. S4), added additional data (Table 1) and have rewritten several paragraphs. All changes to the manuscript are highlighted in blue. Paragraphs that have been removed are crossed out.

Reviewer comments:

Reviewer #1 (Remarks to the Author):

The manuscript describes a set of crystallographic structures for *Leishmania donovani* G6PD showing for the first time its N-terminal domain and trying to explain how this domain would modulate the overall enzyme activity. This N-terminal domain is unique in kinetoplastids G6PDs and it is believed to regulate the enzyme activity (function) in the parasite. How kinetoplastid G6PDs are regulated remains an open issue. The structures and enzymatic data from this manuscript will represent a significant contribution to the field, if authors can address the following major concerns.

Major issues:

1. Deletion of N-domain compromises the formation of LdG6PD tetramer in solution. This is clear in the size-exclusion chromatography experiments with LdG6PD60-562 (Fig. 1a). However, deletion of N-domain, or mutation of Cysteine-138 (C138S) doesn't affect the catalytic efficiency of the enzyme, Kcat/Km values in table 1 for LdG6PD, LdG6PDC138S and LdG6PD60-562 are not significantly different. So, the biological function for the N-terminal domain of kinetoplastid G6PD's remains unsolved. Maybe, the N-terminal interacts with a different regulatory protein. That is not new for eukaryotic G6PDs, see the case of human G6PD and p53 interaction. (Jiang P, et al, p53 regulates biosynthesis through direct inactivation of glucose-6-phosphate dehydrogenase. Nat Cell Biol. 2011 Mar;13(3):310-6.)

We thank the reviewer for these interesting comments. It is true that our studies have shown that the unique N-domain is not critical for enzyme activity. However, in our studies, we found by different methods that the N-domains are involved in tetramerization. Furthermore, we observed a reduced catalytic efficiency of the tetramer compared to the dimer, which was evident through an increased K_M for G6P by 40%. Our postulated LdG6PD tetramer supported this phenomenon since the G6P binding site (α 14; P481-L492) is affected in the tetrameric conformation through the interaction between N-domain B' and loop residues T471-D482 of subunit B.

Of course, the N-terminal domain could also play a crucial role in protein-interaction. Besides p53, trypanothione reductase is thought to interact with G6PD (Gosh *et al.*, G6PD and TR interaction protects *L. donovani* from metalloid mediated oxidative stress, Free Radic. Biol. Med., 2017). We enriched the manuscript with a small paragraph in the discussion that points that out (line 328-333):

'Finally, which function the tetramer has in vivo must still be clarified. Although the conditions we tested had no effect on oligomerization, there are many other factors that could influence the dimer–tetramer equilibrium, such as the product to substrate ratio, various other ROS, or protein interactions. For example, p53 is known to regulate human G6PD activity⁴⁰, and in trypanomatids, trypanothione reductase, a direct NADPH-consumer, is thought to interact with G6PD⁴¹. Maybe the unique N-domain is also involved in such protein interactions, but this needs to be investigated in further studies.'

The function of the N-domain has been studied intensively for 15 years – until now it has remained unresolved. In this work, however, we were able to show for the first time the N-domain in atomic detail and that the entire domain moves upon substrate binding.

We agree with the reviewer that it would be very interesting to understand the function of the N-domain in detail. However, this would require intensive studies, including recombinant production of the interaction protein, production of substrates, kinetic analyses, SEC, HDX-MS, crystallization, production of further mutants etc.

2. The new tetrameric organization proposed for *LdG6PD* would benefit from additional biochemical data (cross-link assay, for instance), to confirm expected structure in solution.

Thank you very much for this comment. However, we are confident of our proposed tetrameric organization, because we have crystallized and analysed at least 30 structures (from different batches, different crystallization conditions, with and without ligands). In all of these structures, with the exception of the N-terminal deletion mutant (*LdG6PD*⁶⁰⁻⁵⁶²), which exhibited also a single dimer peak in the SEC profile, this type of tetramer is present. However, we agree that this tetrameric conformation is very interesting, which is why we would like to further investigate this tetramer and the involvement of the N-domains in another publication, as described under major point 1.

3. In monomer B, poor electron density for the loop connecting the N-domain to Rossmann-like domain results in two possible orientations for the N-domain and makes the structural analysis very confusing.

Thank you for this comment and we apologize for the inadequate presentation. We have rewritten several paragraphs (lines 110-120, 157-175, 311-322), optimized Fig. 2 a, and added a scheme (Fig. 2 b and c) for clarity.

Fig. 2 Overview of the *LdG6PD* wt dimer and schematic model of the *LdG6PD* oligomerization. **a** *LdG6PD* wt dimer in complex with NADP(H) (PDBID: 7ZHU) with position P1 of N-domain B (N^B) is shown in ribbon presentation. The β - α domain (D249-K552) of subunits A and B is colored light purple and light blue, respectively. The 'Rossmann-like' domain (D50-I248) of subunits A and B is colored dark magenta and navy blue, respectively. The helical N-domain with helices $\alpha 1$ (~D11-R31) and $\alpha 2$ (~I36-K49) is colored magenta in subunit A (N^A) and cyan in subunit B (N^B). The connecting loop (residues D50-K64 in subunit A and B) between the N- and the 'Rossmann-like' domain, is colored green. Regions with bad or missing electron density are shown in dotted lines (subunit A: D50-S55, Y476-R479; subunit B: R51-E54).

Helix α 14 as part of the G6P binding site, the flexible loop (T471-D482), and important cysteines (C56, C61, C97, C138, yellow) are labelled. NADP(H) (yellow) and G6P (green), and residues are shown as stick models. To mark the G6P binding pocket, G6P is taken from the superimposed structure with bound G6P (ribbons not shown, PDBID: 7ZHZ). **b** Schematic *LdG6PD* dimer and tetramer involving position P1 of N-domain B (N^B). Subunit B is colored blue and N^B of subunit B in position P1 is colored cyan. Subunit A and A' with N-domain A (N^A) and A' ($N^{A'}$) are colored magenta. **c** Schematic *LdG6PD* dimer and tetramer involving position P2 of N-domain B (N^B). Subunit B is colored blue and N^B of subunit B in position P2 is colored green. Subunit A and A' with N^A and $N^{A'}$ are colored magenta (created with BioRender.com).

4. It is not clear enough how the bind of substrates induces the 45° rotation of N-domain. What are the residues and forces involved in this movement?

This is correct and we are fully aware of this issue, but we cannot answer it properly at the moment. The rotation of the N-domain is clearly visible, but other involved regions show increased B-factors or are disordered. However, residues 471-481 of both subunits are rearranged with certainty upon substrate binding, but are not clearly defined by electron density in some structures (Table S2). We now have described in section 'Substrate binding leads to conformational changes and influences tetramer contacts' the involved residues in more detail (lines 137-166) and calculated polder maps of these regions and added a new figure (Fig. S4) to the supplements. Moreover, there are structural differences around the G6P binding site, described at the end of the same chapter (lines 197-209). We suppose that long range forces induce the rotation of the N-domain. Detailed mechanisms explaining the substrate induced rotation need to be investigated in further studies (for example: mutations of involved residues)

Fig. S4 Substrate-induced shift of the N-domains. The dimers of the apo and the NADP(H) complexed structures are shown in the same orientation (PDBIDs: 7ZHT, 7ZHU). The β - α domain of subunits A or B are colored light purple and light blue, respectively. Electron density map ($F_o - F_c$ polder omit map) contoured at 2.5σ around residues D34-L45 from N-domain A and residues T471-R475 of the loop region are shown in black. **a** apo structure: N-domain A (N^A) is colored grey. **b** NADP(H) complexed structure: N-domain A (N^A) is colored magenta. Important residues are shown in stick models. Respective α -helices and β -sheets are numbered.

5. The bind of G6P in a flipped orientation can also be interpreted as an crystallographic artefact.

Thank you for this comment. In the manuscript we showed only three structures with bound G6P (PDBIDs: 7ZHV, 7ZHW, 7ZHZ). However, we obtained many more crystals with bound G6P, all of which showed a similar orientation of the G6P, although the crystallization conditions were different. Moreover, the binding pocket of G6P does not form crystal contacts in our structures, so we assume that it is not a crystallographic artefact. In addition, we calculated a polder map and show the electron density in figure 5 b.

Fig. 5 Active site close-up of *LdG6PD*. Ribbons and residues of the ‘Rossmann-like’ domain are colored navy blue and of the β - α domain light blue. **a** Close-up of the NADP⁺ binding pocket of the *LdG6PD* structures (PDBID: 7ZHU) within the ‘Rossmann-like’ domain. The NADP(H) moiety is colored yellow. Electron density map ($F_o - F_c$ polder omit map) contoured at 4.0σ for NADP(H), is shown in black. **b** Closeup of the G6P binding pocket. G6P is shown in two conformations. In brown, the G6P conformation from the superimposed *TcG6PD* structure (PDBID: 5AQ1) with the pyranose ring pointing towards the NADP(H) moiety (yellow). In green, the G6P conformation of the *LdG6PD* structures (PDBID: 7ZHZ). Electron density map ($F_o - F_c$ polder omit map) contoured at 4.0σ for G6P, is shown in black. Residues of the NADP⁺ and the G6P binding pocket, are shown in stick models, and hydrogen bonds are indicated with black dotted lines or with grey dotted lines for the G6P moiety from the *TcG6PD* structure.

Minor issues:

1. Line 457: “We crystallized the LdG6PD wt dimer and tetramer fraction separately. From the dimer fraction we obtained crystals complexed with and without NADP(H), with G6P, and with both G6P and NADP(H). From the tetramer fraction, we used a crystal complexed with NADP(H) and G6P. From the dimer fraction of the LdG6PDC138S mutant we got crystals complexed with NADP(H) and with both G6P and NADP(H). For the truncation mutant LdG6PD60-562 we obtained crystals in complex with NADP(H) only. A prerequisite for well-diffracting crystals (< 2 Å) was the presence of both ligands, G6P, and NADP⁺ in the crystallization buffer (Table 2, Table S1).” This would be better in the Results, not in Methods.

We agree with the reviewer. Since, unfortunately, the journal allows only 5000 words for results and discussion we decided to keep this information in the method section.

2. Line 422: “The enzymes remained stable for at least 10 days when stored at 4 °C and 3 months when stored at -80 °C with the addition of 250 mM AmSO₄, as verified via enzyme activity measurements.” Same statement in line 394 and 395

Thank you very much for noticing that. We deleted the statement from line 392-395 (now lines 414-417).

3. Line 379: Please, provide the codon-optimized sequences in the supp. Material. This is an important information to allow data reproducibility.

Thank you for this objection. We added the codon-optimized sequences of *LdG6PD* and *LmG6PD* to the supplemental material as suggested (Table S2).

4. Line 103: which residue atoms were used to calculate the RMSD? Be more specific.

The superimposition was done with coot, using the routine SSM superpose. This is a topology-based alignment. Due to the limited space in the manuscript, we cannot list the residues in detail, but we rephrased this in the manuscript (lines 102-104).

‘Similar to other G6PDs of this family, the LdG6PD dimer (Fig. 2 a) adopts the canonical G6PD fold with RMSD values for the dimer core domain of 2.6 Å and 2.1 Å to human (HsG6PD; PDBID: 2BH9) and Trypanosoma cruzi G6PD (TcG6PD; PDBID: 5AQ1)26–29, respectively (Fig. S3).’

5. Line 111: Replace connected by related

Thank you for that. We have rewritten this paragraph, which is why this sentence does not longer exist.

6. Line 142: Add an image (to supp. Info) of the electron density for these residues.

We calculated polder maps and added additional figures (Fig. S4) to the supplemental material showing the electron density of the contact region between the N^A-domain and subunit B. The interactions between these residues change upon substrate binding, which is now also evident via the electron density map. See also major point 4.

7. Line 103: indicate which residue atoms (C α , mainchain, all ...) were used to calculate the RMSD? Be more specific.

Please see minor point 4.

8. Line 465: Which Pilatus? Be more precise.

We apologize for the missing information and have added this information to the manuscript (lines 486-488).

'Diffraction data for all crystals were collected at beamline X10SA (Pilatus 6M for the crystals of LdG6PD wt and Eiger2 16M for the crystals of LdG6PD mutants) of the Swiss Light Source in Villigen, Switzerland at 100 K and processed with XDS⁴⁷.'

9. Line 472: Crystal with G6P have a larger (not shortene) a -axis than the APO 7zht. Verify.

As shown in Table 2, in space group C121 the length of the a-axis in structures with bound G6P is at least 5 Å shorter than in structures with bound NADP(H). The apo structure crystalizes in space group P121. However, to be more precise, we rephrased the respective sentence (lines 493-495).

'All structures complexed with G6P or NADP(H) show C121 space group symmetry, but the structures with G6P have an a-axis shortened by about 5 Å.'

10. Line 480: Indicate which residues were in the truncated search model.

We thank the reviewer for this comment and checked the truncated model. Indeed, the number of the truncated residues was wrong. We have corrected this in the manuscript (lines 499-504).

'A search model of LdG6PD was generated via homology modeling with SWISS-MODEL⁴⁸ by using HsG6PD (PDBID: 5UKW) as at template. The modelled LdG6PD monomer comprises residues 70 to 549. In spite of the high sequence-identity of 53% (Fig. S3), attempts to solve the structure of the apoenzyme using the canonical G6PD dimer were unsuccessful. Instead, a truncated version of the modelled LdG6PD monomer was used, where loops on the surface, comprising residues 170-179, 359-376, 471-480 and 546-549, were deleted.'

11. Line 480: "The initial Rfree value after the first refinement was 37%. The asymmetric unit of the final solution contained 4 monomers, which form two canonical G6PD dimers. In contrast to other G6PDs, LdG6PD contains an additional N-terminal domain, which was only visible in one dimer of the apo structure." Move this to results.

We agree with the reviewer. Since, unfortunately, the journal allows only 5000 words for results and discussion we decided to keep this information in the method section.

12. Line 488: "...the apo structures". I counted only one APO structure, 7zht. Is that right?

Yes, that is right. Thank you for this comment. We rephrased the respective sentence (line 511).

13. Line 503: replace 7Z4X by 7zht

Thank you, this has been changed (line 526).

14. Line 504: replace 7Z4Y by 7zhy and 7Z4Z by 7zhz.

Thank you, this has been changed (line 527).

Reviewer #2 (Remarks to the Author):

In this manuscript the authors report the first structure of the glucose-6-phosphate dehydrogenase from *Leishmania* parasites, a major tropical disease for which only a few drugs are available and new resistant strains against those drugs are appearing. The authors solved 7 structures of LdG6PD and demonstrated the importance of quite unique N-terminal domain in forming tetramers completely different from those of G6PD enzymes from human and *Trypanosom cruzi*. Without the N-terminal domain, LdG6PD can form the canonical tetramers as in human and Tc G6PDs. Several G6P bound LdG6PD structures indicate that the bound GDP is flipped by 180 degrees around the phosphate positive relative to the conventional G6P binding pose, which is very intriguing. Upon binding of G6P the N-terminal two-helix domain rearranges their orientations relative to the core domain, an interesting observation that might be helpful in designing inhibitors unique to *Leishmania* G6PD without affecting human G6PD.

The extensive enzyme kinetics with a number combinations of the substrates and cofactors largely corroborate the structural results, and the authors proposes a Theorell-Chance mechanism in which G6P binds first followed by NADP⁺ binding, leading to product 6PL and spent cofactor NADPH. The manuscript is overall well written with comprehensive descriptions of the novel G6PD tetramer structures including N-terminal domain shifts upon G6P binding. This is a significant step forward in understanding this class of G6PD and provides important insights into possible inhibitor designs against this neglected tropical disease. It is recommended for publication after the following issues are addressed.

Major points:

- 1. G6P binding pose is flipped by 180 degrees compared with the homologue structures, for example TcG6PD. This is a major departure from the conventional understanding of the G6PD catalysis where the pyranose ring is acted on by the enzyme leading to the production of 6-phosphogluconolactone, hence the pyranose ring needs to be close to the nicotinamide group of NADP(H).**
 - a. Electron density of the unique, unusual G6P pose should be shown.**

Thank you for this objection. We calculated a polder map and show the electron density in figure 5 b. For consistency we included also a figure showing the electron density around NADP(H) (Fig. 5a).

At the beginning of the refinement, we tried to refine the *LdG6PD* wt structure with bound G6P (7ZHV) in its normal position, but then positive difference density ($> 3 \sigma$) could be seen in the channel and negative around the pyranose ring, thus we had to flip G6P. The unusual G6P position was also the best explanation for the electron density in the higher resolution structure of *LdG6PD*^{C138S} (7ZHZ).

Fig. 5 Active site close-up of *LdG6PD*. Ribbons and residues of the ‘Rossmann-like’ domain are colored navy blue and of the β - α domain light blue. **a** Close-up of the NADP⁺ binding pocket of the *LdG6PD* structures (PDBID: 7ZHU) within the ‘Rossmann-like’ domain. The NADP(H) moiety is colored yellow. Electron density map ($F_o - F_c$ polder omit map) contoured at 4.0σ for NADP(H), is shown in black. **b** Closeup of the G6P binding pocket. G6P is shown in two conformations. In brown, the G6P conformation from the superimposed *TcG6PD* structure (PDBID: 5AQ1) with the pyranose ring pointing towards the NADP(H) moiety (yellow). In green, the G6P conformation of the *LdG6PD* structures (PDBID: 7ZH2). Electron density map ($F_o - F_c$ polder omit map) contoured at 4.0σ for G6P, is shown in black. Residues of the NADP⁺ and the G6P binding pocket, are shown in stick models, and hydrogen bonds are indicated with black dotted lines or with grey dotted lines for the G6P moiety from the *TcG6PD* structure.

b. The authors describe that K254, Q440 and A483 make hydrogen bonds with the pyranose ring of G6P. To show the importance of this G6P waiting position, mutations need to be made on K254 and Q440 and enzyme kinetics measured. This will, hopefully, show the significance of the unusual G6P binding pose.

We thank the reviewer for this suggestion. As mentioned in the manuscript (lines 197-209), comparison with homologous structures in which the pyranose ring of G6P points toward the NADP(H) moiety revealed that the conserved residues K254 and Q440 bind the phosphate group. Moreover, in human G6PD, mutation of the corresponding residue K254 showed, that this residue is essential for catalysis (Kotaka *et al.*, Structural studies of G6P and NADP⁺ binding to human G6PD. Acta Crystallogr., Sect. D: Biol. Crystallogr., 2005; Bautista *et al.*, Human G6PD. Lysine 205 is dispensable for substrate binding but essential for catalysis, FEBS Lett., 1995). The importance of a homologous glutamine residue to Q440 for catalysis has also been shown in several studies (Mercaldi *et al.*, The structure of a *T. cruzi* G6PD reveals differences from the mammalian enzyme. FEBS Lett., 2016; Kotaka *et al.*, Acta Crystallogr. Sect. D: Biol. Crystallogr., 2005). We therefore assume that mutation of these residues would result in an inactive enzyme. We know that region D482-A483 and Y251 block the usual G6P binding site when only NADP(H) is bound. However, when G6P is bound in the waiting position, the G6P binding site is open and catalysis could occur. Unfortunately, the exact function of this position is currently unclear and an interesting point for further studies, e.g. with G6P analogues and at transition states.

2. This also raises an important question whether the relationship between the flipped G6P binding and the significant structural changes observed in the N-terminal domains in the tetramer is physiologically relevant to the reaction mechanism. What if G6P sits in the “normal” position with the pyranose ring facing towards the nicotinamide ring of NADP(H)? Would the same set of substantial changes be observed in the N-terminal domain orientations relative the core domains?

We thank the reviewer for this interesting question but we cannot answer it properly at the moment. Binding of only NADP(H) induces a similar rearrangement of the N-domain as seen upon binding of G6P or both substrates. Therefore, we assume that G6P in its "normal" position would also lead to a rotation of the N-domain.

We added the following sentence (*Lines 137-140*):

‘Comparing the apo structure (Fig. 3 a, 4 a-b) with the structures complexed with G6P, NADP(H) or both substrates (Fig. 3 b, 4 c-d) reveals a rotation of the N-domain by about 45° upon ligand binding. In contrast to this striking realignment of the N-domains, only marginal differences in the position of the N-domains are evident between the ligand-bound structures.’

In all structures the rotation of the N-domain is clearly visible, but other involved regions show increased B-factors or are disordered. However, residues 471-481 of both subunits are rearranged with certainty upon substrate binding, but are not clearly defined by electron density in some structures (Table S2). We now have described more details in section ‘Substrate binding leads to conformational changes and influences tetramer contacts’. We calculated polder maps of these regions and added a new figure (Fig. S4) to the supplements. Moreover, there are structural differences around the G6P binding site, described at the end of the same chapter (lines 197-209). We suppose that long range forces induce the rotation of the N-domain. Detailed mechanisms explaining the substrate induced rotation need to be investigated in further studies (for example: mutations of involved residues).

Fig. S4 Substrate-induced shift of the N-domains. The dimers of the apo and the NADP(H) complexed structures are shown in the same orientation (PDBIDs: 7ZHT, 7ZHU). The β - α domain of subunits A or B are colored light purple and light blue, respectively. Electron density map ($F_o - F_c$ polder omit map) contoured at 2.5σ around residues D34-L45 from N-domain A and residues T471-R475 of the loop region are shown in black. **a** apo structure: N-domain A (N^A) is colored grey. **b** NADP(H) complexed structure: N-domain A (N^A) is colored magenta. Important residues are shown in stick models. Respective α -helices and β -sheets are numbered.

3. Disulfide bond between C138 and C56 (line 201-202) in the N-terminal truncated mutant structure, LdG6PD⁶⁰⁻⁵⁶². Normally readers would assume that this was observed in the N-terminal truncated LdG6PD which is mentioned in the sentence immediately preceding this sentence. How is it possible to see a disulfide bond with cysteine C56 which is not included in the construct? The authors need to include a panel or inset in Figure 2 that shows the electron density for the disulfide bond between C138 and C56. Or, is it possible that the disulfide bond is visible in the structure of the WT G6PD in complex with NADP(H) since the sentence refers to Figure 2? If so the authors need to clarify that and rephrase the paragraph.

We thank the reviewer for this observation and apologize for this error. We actually meant the *LdG6PD* wt structure with bound NADP(H). We have rephrased this sentence (lines 221-224).

‘Moreover, the LdG6PD wt structure, complexed with NADP(H), revealed a disulfide bond between C138 and C56; however, with minor conformational changes, an alternative disulfide bond between C138 and C61 or C56 and C97 is also possible (Fig. 2 a). Therefore, we generated a second mutant by replacing C138 with a serine (LdG6PD^{C138S}).’

Minor points:

1. “Pyran” vs. “pyranose”. The authors use “pyran” throughout the manuscript referring to the pyranose ring of G6P. Normally a pyran ring has two double bonds in its 6-membered ring (5C+1O) and close to (but not perfectly) planar conformation while the 6 membered ring of G6P does not have double bonds, hence in the chair conformation. The authors should either clarify that they have indeed the pyran ring in G6P in their X-ray structure, or, use “pyranose” throughout the text.

Thank you, you are right. We changed ‘pyran’ to ‘pyranose’.

2. Figure 2B. Important loops D50-K64 and T471-D482 are shown in both dimers, but according to Table S2, they are only visible in select monomers and dimers. While it is OK to show them in both, the authors should clarify that these were modeled in from the corresponding monomer/dimers.

We thank the reviewer for this remark. We suppose that the reviewer means Table S1 ‘Crystallographic Data’ and figures 3a and 3b ‘Overview of the *LdG6PD* wt tetramer’. It is correct that loops D50-K64 and T471-D482 are only visible in select monomers and dimers. The dotted lines connect the endpoints between the visible residues and indicate missing residues. We improved the figures and rephrased the legend.

Fig. 3 Overview of the *LdG6PD wt* tetramer. The tetramers of the apo **a** and the NADP(H)-complexed structures **b** are shown in the same orientation. The β - α domain (D249-K552) of subunits A/A' and B/B' is colored light purple and light blue, respectively. The 'Rossmann-like' domain (D50-I248) of subunits

A/A' and B/B' is colored dark magenta and navy blue, respectively. **a** apo structure. The helical N-terminal domain with $\alpha 1$ (~D11-R31) and $\alpha 2$ (~I36-K49) is colored grey in subunit A and A' (N-domains N^A and N^{A'}) and colored dark cyan in subunit B and B' (N-domains N^B and N^{B'} with position P1). Dotted lines indicate regions with bad or missing electron density (subunit A/A': G52-V63; subunit B/B': K49-K62, T474-D477). **b** NADP(H)-complexed structure. N-domains N^A and N^{A'} are colored magenta and N-domains N^B and N^{B'} (position P1) are colored cyan. The connecting loop (D56-K64) is colored green, and dotted lines indicate regions with bad or missing electron density (subunit A/A': D50-S55, Y476-R479; subunit B/B': R51-E54). NADP(H) (yellow) and G6P (green) and residues are shown as sticks models. Important residues of the flexible loop (T471-D482) and the connected helix $\alpha 14$ (P481-L492) are directly or indirectly affected by the N-domains and marked with labels or arrows. The apo structure and the NADP(H)-complexed structure were used (PDBIDs: 7ZHT, 7ZHU), and G6P is taken from a superimposed structure with bound G6P (ribbon not shown, PDBID: 7ZHZ).

3. "wt" on line 55: define "WT" in the first occurrence.

Thank you, this has been changed (now line 54).

4. Lines 180-182 on the G6P interactions with G6PD. This sentence reads as if it is referring to the LdG6PD structure. But actually, it is about the G6P pose and interactions in the TcG6PD structure. This should be explicitly stated in the sentence.

Thank you for this objection, the paragraph was indeed confusing and has now been rephrased (lines 197-201).

'Comparison with homologous structures, in which the pyranose ring of G6P points towards NADP+, revealed a corresponding G6P binding pocket in LdG6PD, which is formed by residues K220, D249-Y251, F286-E288, K406, and K411-V413 (Fig. 5 b). According to homologous structures, the pyranose ring of G6P would be coordinated by residues K220, E288, I304, D307, H312, and K406, and the phosphate moiety would be bound by residues H250, Y251, K254, K411 and Q440.'

5. Lines 217-218. This sentence is not precise, or almost incorrect. Actually, the WT tetramer coordinates are generated (constructed) by a different symmetry operation of totally different space groups, C121. As for 7ZHT, according to Table S1, it has already two dimers in the AU, so the tetramer might be exactly as shown in Figures 2 and 3; if not it can be generated via a P121 symmetry operation.

We thank the reviewer for this comment. The LdG6PD wt and the LdG6PD^{C138S} mutant complexed with NADP(H) and/or G6P adopt C121 space group symmetry, and via crystallographic symmetry operation the same kind of tetramer can be formed (see figure 3b). The apo form crystallizes in P121 with two dimers in the AU, however these two dimers do not form the biological tetramer (Fig. 3), they are quasi two half tetramers. Via P121 symmetry operation one can construct the same kind of tetramer (Fig. 3a) as constructed for the LdG6PD wt and the LdG6PD^{C138S} mutant (Fig. 3b), except the position of the N-domains. Regarding the N-terminal truncation mutant LdG6PD⁶⁰⁻⁵⁶² with one monomer in the AU and C222 space group symmetry, this type of tetramer cannot be formed via C222 symmetry operation, instead we can construct another type of tetramer (Fig 6c). We rephrased that paragraph (line 232-239) and the construction of the tetramer (line 110-120).

6. Figure S4: define the “nine” sequences. The figure shows only 5.

Thank you very much for noticing that. That was a mistake. We rephrased the respective sentence in the figure legend. Figure S4 is now Figure S3.

‘Strictly conserved residues are highlighted with black background, while highly similar residues, with similar physicochemical properties in at least four out of the five sequences are shown in bold letters.’

7. Finally, it would be interesting and instructive to discuss possible strategies for designing drugs against Leishmania parasites taking advantages of the unique N-terminal domain and its interactions with the active sites across the loop T471-D482 or D50-K64.

This is indeed interesting, which is why we already started with docking studies and inhibitor testing. However, such intensive studies would exceed the scope of this manuscript, which is why we decided to address this in another publication.

2. Revision – 10/31/2022

ANSWERS TO THE COMMENTS OF THE REVIEWERS

Manuscript ID: COMMSBIO-22-1550-A

Title: Chasing the Tail: Crystal Structure of Leishmania donovani Glucose 6-Phosphate Dehydrogenase Reveals a Unique N-terminal Domain

The authors wish to thank the reviewers for their constructive comments and for giving us the opportunity to resubmit our work. We updated the discussion to clarify the key statements of our manuscript. All changes to the manuscript are highlighted in yellow.

Reviewer comments:

Reviewer #1 (Remarks to the Author):

Despite answering all raised issues, the reported data is not robust enough to rebut the argument that the tetrameric arrangement proposed for LdG6PDH could be a crystallographic artefact. The electron density map presented in Fig5. B doesn't have enough resolution to allow the accommodation of the G6P in this unusual, flipped orientation. Maps from higher resolution structure would be necessary to convince article readers that the flipped G6P is correct. The reported difference (of 40%) between Km-G6P values for LdG6PDH dimer and tetramer is too small to believe that changes between dimer and tetramer would have any biological significance for the parasite growth or viability. The new finds based on Leishmania G6PDH structures go against what was established based on T. cruzi and Human G6PDH structures. So, these finds require more and better data to be trustable.

To date, there are only 29 G6PD X-ray structures from two prokaryotic and three eukaryotic species (*Homo sapiens*, *Trypanosoma cruzi* and *Kluyveromyces lactis*) available. The three previous published *T. cruzi* G6PD and the three tetrameric human G6PD structures show a maximal resolution of 2.65 Å. In our current study, we have measured more than 30 LdG6PD crystals with and without substrates up to a resolution of 1.7 Å. Furthermore, our studies have been carried out with wild type and mutated enzyme.

Previous studies on *T. cruzi* G6PD have postulated a tetramer with a back-to-back orientation of the two dimers. This back-to-back arrangement was also found in the *human* G6PD tetramer, but here the dimers are oriented differently than in *T. cruzi*. Therefore, the tetramers of the two species are not identical. This is not surprising, since trypanosomatid G6PDs, such as from *Trypanosoma* and *Leishmania*, contain a unique N-terminal domain, which is not present in the human enzyme. Previous crystallization studies of *T. cruzi* G6PD were performed using only a truncated mutant lacking the N-terminal domain. If we also truncate the N-terminal domain in *LdG6PD*, we are able to construct via crystallographic symmetry operations the same tetramer arrangement as seen for *T. cruzi* G6PD. However, the tetramer peak is then no longer visible in the SEC profiles of the truncated *LdG6PD*, in contrast to full-length *LdG6PD* where a dimer and a tetramer peak are observed in the SEC profiles. In our studies crystallization of full-length *LdG6PD* always reveals a face-to-face tetramer in which the N-domains are involved in tetramer formation.

It was previously postulated, that all *Kinetoplastida* G6PDs would form a tetramer via conserved salt bridges. In contrast to the studies on *T. cruzi* G6PD, our study proved that the dimer-tetramer equilibrium of untruncated *LdG6PD* is independent of ionic strength, suggesting a minor contribution of salt bridges in *LdG6PD* tetramer formation. Therefore, we suppose that the *T. cruzi* back-to-back tetramer is an artefact, due to the truncation of the N-terminal domain.

For the above reasons, it cannot be assumed that previous knowledge about *Trypanosoma* G6PDs is 'established' and could therefore be used as a standard or template. Despite intensive studies over the last 15 years, the role of the N-domain has not yet been definitively clarified. Our studies provide a first and very important approach to this question. Similarly, all studies in the field of G6PD also covered only the methods we used and no studies have yet elucidated the physiological significance of the postulated dimer and tetramer conformations of G6PDs.

Finally, we would like to emphasize that the tetrameric arrangement of *LdG6PD* is not the main statement of our manuscript. In our work, we present for the first time a G6PD structure from the *Trypanosomatidae* family that contains the complete N-domain. Moreover, we show that this domain changes its position relative to the main domain upon substrate binding, and together with our kinetic studies, we gained new insights into the binding mode of *LdG6PD*.

Although we agree, that investigating the function of the tetramer in solution would be very interesting, this would certainly take at least half a year and is beyond the scope of this publication.

Reviewer #2 (Remarks to the Author):

The authors have addressed all the points raised by this reviewer. The revised manuscript is recommended for publication in Communications Biology.

We are very pleased that we were able to address all of the concerns and that the reviewer fully recommends publication.